

# Regional rainfall thresholds for landslide occurrence using a centenary database

Teresa Vaz[1], José Luís Zêzere[1], Susana Pereira[1], Sérgio C. Oliveira[1], Ricardo A. C. Garcia [1], Ivânia Quaresma[1]

[1] Centre for Geographical Studies, Institute of Geography and Spatial Planning, Universidade de Lisboa, Lisbon, 1600-276, Portugal

*Correspondence to*: Teresa Vaz (tvaz@campus.ul.pt)

**Abstract.** This work proposes a comprehensive methodology to assess rainfall thresholds for landslide initiation, using a centenary landslide database associated with a single centenary daily rainfall dataset. The methodology is applied to the Lisbon region and include the rainfall return period analysis that was used to identify the critical rainfall combination (quantity-duration) related to each landslide event. The spatial representativeness of the reference rain gauge is evaluated and the rainfall thresholds is assessed and validated using the receiver operating characteristic (ROC) metrics.

Results show that landslide events located up to 10 km from the rain gauge can be used to calculate the rainfall thresholds in the study area; however, such thresholds may be used with acceptable confidence up to 50 km distance from the rain gauge. The obtained rainfall thresholds using linear and potential regression have a good performance in ROC metrics. However, the intermediate thresholds based on the probability of landslide events, established in the zone between the lower limit threshold and the upper limit threshold are much more informative as they indicate the probability of landslide event occurrence given rainfall exceeding the threshold. This information can be easily included in landslide early warning systems, especially when combined with the probability of rainfall above each threshold.

## 1 Introduction

Rainfall is the most important physical process for the landslide triggering in Portugal (Zêzere et al., 2015; Vaz and Zêzere, 2016) as worldwide (e.g. Crozier, 1986; Crosta and Frattini, 2008). However, the relation between rainfall and landslides is indirect and typically include a process cascade where the rainfall is followed by the infiltration into the soil, which increases the pore-water pressure that is responsible at last by the decrease of the slope materials shear strength (Terlien, 1998; Glade and Crozier, 2005).

During the last decades, the relationship between landslides and rainfall has been tentatively established by the assessment of rainfall thresholds, i.e., rainfall conditions (quantity, intensity) that when reached or exceeded can induce a landslide event (Reichenbach et al., 1998; Guzzetti et al., 2007). The rainfall thresholds for slope failure have been proposed following the physical and the empirical approach. The first approach considers the physical basis of the process, using hydrological models and stability calculations (Terlien, 1998; Iverson, 2000; Frattini et al., 2009). However, it demands high resolution



data (e.g. groundwater conditions; shear strength properties) that often are not available for large areas (Guzzetti et al., 2007). The second approach is statistically-based and is sustained by historical records regarding landslide events and rainfall data series (Guzzetti et al., 2007). Several thresholds have been proposed worldwide using the empirical approach that can differ according to the kind of rainfall measurement and the number of rain gauges used to calculate the threshold, as well as the geographical extend where the threshold is applied.

The most common empirical rainfall thresholds used at the local and regional scale are the rainfall intensity and duration (I-D) threshold, the event-duration (E-D) threshold, the antecedent rainfall threshold, and the combined threshold. The I-D threshold link the total depth of rainfall and the instantaneous rainfall intensity (Caine 1980) and has been widely used as a power-law threshold (e.g. Guzzetti et al., 2008; Saito et al., 2010; Brunetti et al., 2010). It has been applied with good performance especially for shallow landslides, triggered during periods of rainfall of short duration. Alternatively to the former, the E-D threshold associates the rainfall cumulative event and the rainfall event duration (e.g. Peruccacci et al., 2012). The antecedent rainfall thresholds assess the influence of the antecedent precipitation on the groundwater levels and soil moisture, thus acting as landslide preparatory factor. This is particularly important for deep-seated landslides induced normally by long-lasting rainfall periods (Martelloni et al., 2012). However, the definition of the critical rainfall period for the antecedent rainfall is an important source of bias (e.g. Guzzetti et al., 2007; Zêzere et al., 2015) and different periods have been proposed in the literature ranging from few days to several months (e.g. Glade et al., 2000; Cardinali et al., 2006). Finally, the combined thresholds include several combination as the rainfall event combined with rainfall intensity (e.g. Onodera et al., 1974); the event rainfall with the antecedent rainfall (e.g. Pereira and Zêzere, 2012); the event rainfall with the antecedent calibrated rainfall (e.g. Zêzere et al., 2005).

The rainfall thresholds for landslide activity obtained in a study area cannot be extrapolated for other regions, namely because of changes regarding the climatic regime (Glade et al., 2000). To allow the comparison of rainfall thresholds obtained in different areas, the normalization of rainfall data has been made using two climatic indices: the mean annual precipitation (MAP) (Cannon, 1988) and the rainy day normal (RDN) (Wilson, 1997). The different rainfall parameters can be divided by the two climatic indices to obtain, for instance, the normalized intensity-duration (e.g. Wieczorek et al., 2000), the normalized event-duration (e.g. Giannecchini, 2005) and the normalized antecedent rainfall (e.g. Aleotti, 2004).

The rainfall measurements used to assess rainfall thresholds for landslide activity can be based on a single reference rain gauge (e.g. Zêzere et al., 2005; Marques et al., 2008; Martelloni et al., 2012) or on multiple rain gauges (e.g. Peruccacci et al., 2012). The near distance, similar elevation and topographical and morphological settings are pointed as the preferable criteria to select the representative area of a rain gauge (Brunetti et al., 2010). However, the distance to where the rain gauge is spatially representative is a critical point that often is not addressed, which can be an additional source of bias for the threshold definition.

The assessment of rainfall thresholds implies the consideration of two types of information, liking rainfall and landslides in a single study area: the rainfall events that triggered landslides in a defined time period in the past; and the rainfall events that did not trigger landslides in the same time period. Considering the rainfall data sets associated (and non-associated) with



landslide events two distinct rainfall thresholds can be defined: (i) the lower limit threshold, which is the limit below which the landslides have not been recorded; and (ii) the upper limit threshold, which is the limit above which landslides have always been recorded (Glade et al., 2000). The zone between the lower limit and upper limit thresholds includes rainfall conditions that triggered and did not trigger slope failures in the past. As a rule, the uncertainty increases with the increasing gap between the lower limit and upper limit thresholds. Therefore, between the lower threshold and the upper threshold different probabilities of landslide occurrence exist that are important to quantity.

The main purpose of this study is to present and discuss a comprehensive methodology to assess rainfall triggering thresholds, using a centenary landslide database associated with a single centenary daily rainfall dataset. In addition, five specific objectives are stated: i) to identify the critical rainfall quantity-duration combinations for landslide occurrence; ii) to compute the antecedent rainfall thresholds using linear and potential regression and defining the lower limit and the upper limit rainfall thresholds; iii) to assess the thresholds performance using receiver operating characteristic (ROC) metrics; iv) to estimate the probability of rainfall threshold and the probability of landslide events above a specific rainfall threshold; and v) to identify the geographical area where the rainfall thresholds can be applied.

## 2 Study area and general characteristics of the precipitation regime

The Lisbon region is located in the southern Portuguese Estremadura, being divided in two parts by the Tagus River (Fig. 1). The landscape is marked by hills and valleys and three mountains of limited extension and altitude (Fig. 1): the Montejunto Mountain at the Northwest (666 m altitude), the Sintra Mountain at the West (528 m) and the Arrábida Mountain at the South (501 m).

The climate in the Lisbon Region, as in Portugal, is influenced by the subtropical anticyclone and the sub-polar depression zone ( Espírito Santo et al., 2014; Lima et al., 2015). The atmospheric general circulation combined with the orography and the oceanic and continental influences are the most important factors that shape the regional climate (Nunes and Lourenço, 2015). The precipitation regime is typically irregular, with an inter-annual and intra-annual variability (Kutiel and Trigo, 2014). The inter-annual variability is notorious in the centenary annual precipitation data registered at the Lisboa-Geofísico rain gauge (Fig. 2). The mean annual precipitation (MAP) is 709 mm, but the variability is very high and wet years can be followed by severe dry years. In some climatological years the annual precipitation reached the double of MAP (e.g. more than 1400 mm in 1876/77), while other climatological years did not reach half of the MAP (e.g. less than 300 mm in 2004/05).

The intra-annual precipitation regime is characterized by the seasonality (Fig. 3), with two important seasons (dry and wet) separated by transition periods (Ribeiro et al., 1999). During two months of summer (July and August) the precipitation is almost absent in quantity and frequency. On average, these months concentrate only 1.3 % of the annual precipitation. The Azores anticyclone influence, in its north-westerly position, explains the warm and dry air that affect the Lisbon region during this season (Trigo and DaCamara, 2000). The monthly precipitation is highest from October to March, however with





a strong inter-annual variability. On average, this period concentrates more than 75 % of the annual precipitation, with a frequent peak in November. This wet period is explained by the large-scale circulation led by the Icelandic low pressure system, which bring moist air responsible for rainfall events (Trigo and DaCamara, 2000). September (April, May June) are transition months between the dry (wet) and wet (dry) seasons and can be highly variable from one year to the another

concerning the amount of rainfall.

As a rule, the circulation weather types, associated with high precipitation amounts, are of cyclonic and westerly type (Trigo and DaCamara, 2000; Ramos et al., 2014). Recently, it was found that the winter storms in Europe, responsible for large accumulations of precipitation, have a tendency for temporal cluster (Mailier et al., 2006; Vitolo et al., 2009; Pinto et al., 2013). Therefore, storms with high magnitude are followed by other storms, increasing the probability to induce other

natural hazards, as floods and landslides.

## 3 Methodology

### 3.1 Identification of landslide events, selection of rain gauge and identification of critical rainfall combinations

The landslide database used in this study includes the DISASTER database and has detailed information about the date and location of landslides occurrence. The DISASTER database was carried out exploring several daily and weekly newspapers, published between 1865 and 2010, and includes all the landslides that caused fatalities, injuries, missing people, evacuated

and homeless people. The methodology used to construct the DISASTER database has been widely described and can be found in Zêzere et al. (2014). Additionally, using the same newspaper sources, landslides that did not caused any human damage during the same time period were identified and included in the database that supported this study. Our analysis is focused on the date of landslide occurrences. So, the newspapers are a reliable data source, despite the existing uncertainty

concerning the spatial location of many reported landslide events and their typology. Likewise, the landslides typology is often unknown. Therefore, similar to other studies (e.g. Brunetti et al., 2010; Rosi et al., 2012; Peruccacci et al., 2017;), the analysis was performed for all landslides typologies.

Along this study the following definition was adopted for landslide event: an individual landslide or a set of landslides that occurred on a precise date (day). In those cases where the activity period of a landslide was reported as lasting during several

days, the first day of the period was considered for the landslide event.

The selection of the reference rain gauge took into account the available time series, the data quality and resolution and the climatic representativeness. The daily rainfall data was collected at the Lisboa-Geofísico rain gauge (Latitude 38.72 °N, Longitude 9.15 °W, Elevation 77 m), located within the Lisbon city. The rainfall daily measurements at Lisboa-Geofísico started in 1864, being one of the few rain gauges with centennial-long daily records in Portugal. A long time series of rainfall

data is an important condition to create comprehensive thresholds that are based on the analysis of the rainfall return period. In addition, this rain gauge presents a reliable data, whose quality and completeness was already tested and confirmed by





Kutiel and Trigo (2014). The rainfall measurements have been made without interruption and always in the same place, since 1864. Furthermore, the rain gauge is climatically representative of the Lisbon region, with a rainfall regime influenced mainly by the atmospheric general circulation and the oceanic proximity.

The daily precipitation refers to the period between 9:00 UTC of the previous day and the 9:00UTC of the day of 5 measurement, whereas the landslides dates are ascribed to a period between the 0:00UTC to 23:59UTC. Due to this difference, the date of each landslide event reported by the newspaper was compared with the daily rainfall registered in three days (starting from the day before up to the day after), and the day registering the highest rainfall amount was selected as the day of the landslide event.

The reconstruction of accumulated rainfall follows the methodology proposed by Zêzere et al. (2005). In a first step, the 10 daily rainfall data registered at the Lisboa-Geofísico rain gauge during the period 1864/65 - 2009/10 were organized by climatological year (September to August). The decision to use the climatological year, instead of the hydrological year (October-September), is justified by the precipitation regime of the study area. Starting the analysis in September, after the month with the low values of precipitation (August), we capture the complete transition period towards the wet season in each year. Afterwards, for each day, from 1864 to 2010 the cumulative antecedent precipitation was calculated for the 15 durations of 1, 2, 3, 4, 5, 10, 15, 20, 30, 40, 50, 60, 75 and 90 days, following the Eq. (1):

$$Px_{(n\ duration)} = Px + Px_{-1} \ldots + Px_{-n} \,, \tag{1}$$

where $Px$ is the daily rainfall for the day $x$, $Px_{-1}$ is the daily rainfall for the day before the day $x$; and $Px_{-n}$ is the daily rainfall for the $n$ days before the day $x$.

20 The maximum annual records of daily rainfall and accumulated rainfall for each duration were extracted and analysed using the theoretical distribution described by Gumbel (Gumbel, 1958). This distribution is also known as the distribution of Ficher-Tippett and is applied for the extreme values. With the Gumbel law it is possible to obtain the probability of occurrence of each precipitation value within the series with $N$ values. The reduced Gumbel distribution ($y$) is calculated with the following Eq. (2):

$$y = -\ln(-\ln \frac{m}{N+1}) \,, \tag{2}$$

Where, $m$ is the position number of observation of the respective observation and $N$ is the total number of observations. Considering this distribution, the theoretical frequencies can be calculated by the average and standard deviation for the reduced Gumbel distribution ($My$ and $Sy$) and for the precipitation values ($Mx$ and $Sx$). The following Eq. (3) expresses the theoretical trend:

$$y = \alpha(x - \mu), \tag{3}$$

Where, $y$ is the reduced variable and $x$ the precipitation value. The parameters $\alpha$ and $\mu$ are calculated as follows:

$$1/\alpha = Sx/Sy, \tag{4}$$



$$\mu = Mx - My/\alpha, \tag{5}$$

Finally, the probability of exceedance of any rainfall value is given by the Eq. (6):

$$P(x) = 1 - e^{-e^{-y}}, \tag{6}$$

For each landslide event the cumulative antecedent precipitation was assessed for the durations of 1, 2, 3, 4, 5, 10, 15, 20, 30, 40, 50, 60, 75 and 90 days. For each antecedent precipitation the return period (RP) was calculated with the following Eq, (7):

$$RP = \frac{1}{1 - e^{-e^{-y}}}, \tag{7}$$

The pair (quantity-duration) with the highest return period was considered as the critical rainfall combination responsible for triggering the landslide event. This assumption is not physically based, but has been applied in previous work (e.g. Marques et al., 2008; Zezere et al., 2008; Zêzere et al., 2015) and provides the best discrimination of the rainfall events related with landslide activity (Zêzere et al., 2005). Moreover, this approach agglomerates the rainfall that triggered the landslide event

and the antecedent rainfall that contributed as landslide preparatory factor.

As it was previously mentioned, our landslide database was collected from newspaper sources and in some cases the rainfall triggering is not clear. Therefore for threshold calculation we decided to use only the landslide events whose critical rainfall combination have a return period exceeding 3 years. The boundary is arbitrary, but this criterion reduces the possibility of considering landslide events whose triggering factor was other than rainfall (e.g. human action). The landslide events

associated with critical rainfall combinations with return period less than 3 years were assumed as not triggered by rainfall.

Finally, the climatological years without landslide records in the database were selected and the maximum yearly accumulated rainfall was identified for durations lasting from 1 to 90 consecutive days. These data were further used as rainfall events that did not generated landslide events and are crucial for the thresholds definition and validation.

### 3.2 Critical distance from the rain gauge

The critical distance where the rain gauge is regionally representative was evaluated by drawing several buffers up to 60 km from the rain gauge (5, 10, 20, 30, 40, 50 and 60 km). The ratio between the non-rainfall triggered landslide events and the rainfall triggered landslide events within each buffer was used to identify the area where the rain gauge is representative.

During the analysed time period (1864/65 - 2009/10) landslides in the study area were mostly triggered by rainfall and the earthquake triggering can be neglected (Vaz and Zêzere, 2016). The human action was an additional landslide triggering

factor, in particular through artificial cuts and drainage constrains associated with the progressive enlargement of urban areas. As it was already mentioned, the reference rain gauge is located in the city of Lisbon, where the landslides induced by human action are expected to be in higher number when compared with the outside of the urban area. Following this





assumption, the ratio between the non-rainfall triggered landslide events and the rainfall triggered landslide events should decreases as the distance from the gauge increases. If this relation does not occur we assume that the rain gauge is not anymore representative for the corresponding buffer. Therefore, the lowest ratio between non-rainfall triggered landslide events and rainfall triggered landslide events was considered to define the critical distance where the rain gauge is regionally

representative to assess rainfall thresholds for landslide occurrence.

### 3.3 Rainfall triggering thresholds assessment and validation

Landslide events registered within the critical distance from the rain gauge were considered and rainfall thresholds were established using linear and potential regression, based on rainfall quantity-duration with the highest return period. The lower limit and the upper limit rainfall thresholds were also defined following the suggestion by Glade et al. (2000). When

representing thresholds we avoid using logarithm scales and thresholds were established as linear relationship instead of power law, with a single exception (the potential regression threshold). These options maximize the zone between the lower limit and upper limit thresholds, thus allowing the distinction between rainfall events that generated (not generated) landslide events.

The performance of rainfall thresholds was evaluated using ROC metrics. ROC analysis are commonly used to validate

susceptibility landslide models (Beguería, 2006; Kappes et al., 2011) and it is based on confusion matrices. The principles used in theses analyses can also be applied to validate the rainfall thresholds (e.g. Staley et al., 2013, Gariano et al., 2015a, Zêzere et al., 2015). The confusion matrix is used to assess the correct and incorrect predicted observations, for positive and negative cases (Beguería, 2006). Therefore, the analysis is based on the evaluation of True Positive (TP), False Negative (FN), True Negative (TN) and False Positive (FP) cases. When, applied to rainfall thresholds the TP correspond to the

landslide events which rainfall combination (quantity-duration) is above the threshold. The FN are landslides events for which the rainfall combination (quantity-duration) is below the threshold. The rainfall combinations that did not resulted in landslides events are classified as TN if they are below the threshold or FP if they are above the threshold.

Also, four ROC metrics functions described by Staley et al. (2013) were used in this study (Table 1). The True Positive rate (TP$_r$) is the proportion of landslide events that were correctly predicted by the threshold (Table 1). The False Positive rate

(FP$_r$) is the proportion of rainfall events above the threshold for which there is no information of landslide occurrence. The False Alarm rate (FA$_r$) is the ratio between false predictions and the complete set of rainfall events above the threshold. The Threat score (TS) is used to evaluate the threshold to maximize the number of correct predictions while minimize the rate of FP and FN. A TS = 1 represents a perfect model, being reduced by the incorrect predictions.

The probability of a rainfall event above the rainfall threshold resulting in a landslide event, was measured by the Positive

Predictive rate (PP$_r$), which was previously described by Bradley (1997) and Fawcett (2006). The PP$_r$ measures the relation between the rainfall events above the threshold that resulted in landslide events and the complete set of rainfall events located above the threshold, as follows:

$$PP_r = \frac{TP}{TP+FP} \ ,$$
(8)




Therefore, the $PP_r$ is the opposite of the $FA_r$, and can also be calculated by the expression:

$$PP_r = (1 - FA_r) \, , \qquad\qquad (9)$$

Using this approach, several linear rainfall thresholds were plot in the zone between the lower limit and the upper limit rainfall thresholds, and the corresponding $PP_r$ were calculated in order to compute the probability of landslide event associated to each threshold. In addition, the probability of each rainfall threshold was computed based on the return period of the corresponding rainfall quantity-duration.

Lastly, the performance of the lower limit threshold was assessed beyond the critical distance of the rain gauge. For each
buffer referred in Sect. 3.2 the ratio between the FN and the total set of landslide events (TP + FN) was systematically evaluated. We assume the lower limit threshold can only be applied to those buffer distances where this ratio remains stable.

## 4 Results

### 4.1 Landslide events and critical distance from the rain gauge

Within the area located up to 60 km-distance from the reference rain gauge 223 landslide events were identified dating from
1865 to 2010 (Fig. 4). The return period computed for the rainfall quantity registered from 1 to 90 consecutive days prior 92 landslide events does not exceed 3 years. Therefore, according to the criterion defined in Sect. 3.1, these landslide events were assumed not to have been triggered by rainfall.

The ratio between the number of non-rainfall triggered landslide events and the rainfall triggered landslide events was calculated for each buffer zone showed in Fig. 4. The results are summarized in Table 2 and were used to define the critical
distance where the rain gauge is regionally representative, and to select the landslide events considered to compute the rainfall thresholds. Within the 5 km buffer the calculated ratio is relatively high (0.65). The first buffer zone includes the Lisbon city centre, which explains the high number of landslides triggered by factors other than precipitation, mainly due to human actions. In the following buffer zone (10 km) the ratio decreases to 0.63. This decrease was expected as the urban area extension decreases in the second buffer, thus justifying the lower number of non-rainfall triggered landslides. The ratio
between the non-rainfall triggered and the rainfall triggered landslide events increases to 0.66 within the 15 km buffer zone, and the ratio range between 0.66 and 0.70 in the next buffer zones up to 60 km distance from the rain gauge. The increasing ratio in distances exceeding 10 km from the rain gauge cannot be attributed to the occurrence of a non-expected high number of non-rainfall triggered landslide events, but can only be explained by the decrease of spatial representativeness of the rain gauge data in areas beyond the 10 km distance. Therefore, we consider the 10 km distance the critical distance where the rain
gauge is representative, and the rainfall thresholds were computed considering only the landslide events registered within this zone.



In the area located up to 10 km distance from the reference rain gauge of Lisboa-Geofísico 96 landslide events were identified, which include 258 individual landslides. The yearly and monthly distribution of these landslide events are shown in Fig. 2 and Fig. 3, respectively. The landslide events occurred mainly in wet years: 89 % of total landslide events were registered in years with precipitation above the mean annual precipitation (MAP). The climatological years 1876/77,

1946/47 and 1968/69 were the most relevant in number of landslide events (6 events each of them). In these three climatological years the annual rainfall was above 933 mm at the reference rain gauge, which exceeds the MAP more than 30 %. However, there is not a direct relationship between the MAP and landslide events because landslide occurrence is usually related with rainfall events occurred in a few days or weeks, which are not expressed by the mean annual precipitation. Indeed, landslide events were also registered in eight years with annual precipitation below MAP, as was the

case of 1909/10 that was characterized by the occurrence of two landslide events.

The monthly distribution of landslides events follows the rainfall distribution along the year in a Mediterranean climate, with dry summers and wet winters. The landslide events are essentially coincident with the rainiest months, as 92 % of events occurred from November to March. Within this period, the months of January and February stand out with the highest concentration of landslide events (24 % and 22.9 %, respectively). Besides the monthly rainfall percentile, the Fig. 3

represents the 30-day cumulative antecedent rainfall for each landslide event and points out that 96 % of landslide events are above the 70 percentile. If we consider the 90 percentile this value decrease to 79 %, but it continues to highlight the exceptionality of rainfall during the 30 days before the triggering of landslides.

For each landslide event the critical rainfall quantity-duration was obtained following the methodology described in Sect. 3.1. The obtained critical durations associated with landslide events range from 1 to 90 consecutive days. The relationship

between critical duration and month is showed in Fig. 5 for the rainfall-triggered landslide events. The shorter durations rainfall events (less than 20 consecutive days) occurred mainly from September to December (56 %), in the beginning of the rainy period. On contrary, when associated with longer durations rainfall periods (more than 20 consecutive days) the landslides events were more frequent from January to May (86 %).

Figure 6 represents the rainfall quantity-duration combinations that resulted in landslide events and the typical return periods

established for 5, 10, 25, 50, 100, 150 and 200 years. Around 64 % of the rainfall quantity-duration that resulted in landslide events have a return period below 10 years. However, four landslide events had a rainfall amount and duration with a return period very high, above 150 years. The Fig. 6 also detaches the landslide events that include multiple landslides and the landslide events that are constituted by a single landslide. The distribution of both groups is inconclusive, as the landslide events containing multiple landslides are not always directly related with the exceptionality of the rainfall event, i.e., with

critical rainfall quantity-duration combination with higher return period.

## 4.2 Rainfall thresholds for landslide triggering

The rainfall conditions (quantity-duration) associated with each landslide event were considered to define rainfall thresholds using linear and potential regression (Fig. 7). The linear regression follows the equation y = 5.5 D + 124.6, where D is the





duration in days, whereas the potential regression follows the equation: $y = 67.8\ D^{0.46}$ (Table 3). The coefficient of determination ($R^2$) is very high in both cases (0.8 and 0.9, respectively). Both rules can be used as rainfall threshold for landslide occurrence in the study area; however none of them ensure the false negatives occurrence (i.e. landslide events below the threshold).

To validate the thresholds, the maximum yearly rainfall for each duration (1 to 90 consecutive days) was calculated for those climatological years without records of landslide events in the analysed period (1864/65 - 2009/10). These records represent rainfall events not associated with landslides and are symbolized by grey dots in Fig. 7 (102 dots). The majority of these rainfall events (96.8 %) drop below the threshold obtained with the potential regression. However, there are 57 false negatives occurrences (i.e. events that occurred without being predicted), as well as 48 false positives (i.e. rainfall events

lying above the threshold, without any landslide reported).

In the next step, the lower limit and the upper limit rainfall thresholds were determined. The former establish the threshold below which there are no true positives (landslide events), whereas the latter establish the threshold above which there are no false positives (rainfall events without landslides). The lower limit threshold follows the equation $y = 4.4D + 56.5$, and the upper limit threshold follows the equation: $y = 7.3D + 235.8$, where D is the duration in days (Table 3).

Table 4 summarizes the ROC metrics for the regression thresholds (linear and potential) and the lower limit and the upper limit thresholds. The $TP_r$ measure the proportion of landslides events that occurred when the combinations of rainfall-duration are exceeded and shows the efficiency of a threshold to predict a landslide event. On the other hand, the $FP_r$ measures the proportion of combinations of rainfall-duration that are above the threshold but did not result in any known landslide event. For the potential regression threshold, the $TP_r$ is not very high (0.41, best value is 1) but the represents good

results with a low value (0.03, the best value is 0) which means that the threshold have a low probability of a false detection. The $TP_r$ is equal to 1 for the lower limit threshold, considering that it was drawn to avoid FN occurrences. However, the $FP_r$ and the $FA_r$ are very high (0.37 and 0.85, respectively) as consequence of the typical low values of the threshold. The lower limit is a conservative threshold, whose main advantage is predicting all the landslide events, but also including a very high number of false events. On the contrary, the upper limit thresholds is only surpassed by true positives occurrences, so the $FP_r$

and $FA_r$ have the best result (0 value); however, the $TP_r$ is very low (0.03) reflecting the high number of false negative events. The Threat score (TS) provides a better understating of each threshold performance as it relates the TP, FN and FP occurrences. The linear regression threshold has the best result with 0.29 of TS, when compared with the potential regression threshold (0.27), the lower limit (0.15) and the upper limit (0.03) thresholds (Table 4). The False Alarm rate ($FA_r$) also gives a better result for the linear regression threshold in comparison with the potential regression threshold (0.47 and 0.55,

respectively).

Therefore, regression thresholds, linear or potential, can be used as acceptable thresholds to predict landslide events in the study area. However, the lower limit and the upper limit thresholds should not be excluded, as the zone between these rainfall thresholds defines the boundary conditions where any rainfall event may (or may not) originate a landslide event.



### 4.3 Probability of landslide event and probability of rainfall above the threshold

The $PP_r$ summarized in Table 4 gives the probability of a rainfall event resulting in a landslide event when the threshold is exceeded. The value ranges from 0 to 1 where 1 indicates 100 % probability of landslide occurrence. Accordingly, when the lower limit threshold is exceeded, the probability of occurrence of a landslide event is relatively low (0.15). On contrary,

when the upper limit threshold is reached the occurrence of a landslide event is certain ($PP_r = 1$). The $PP_r$ associated with the regression thresholds is close to 0.5, being higher for the linear trend in comparison with the potential trend (0.53 and 0.45, respectively).

The systematic comparison between True positives and False positives and the $PP_r$ calculation were taken into consideration to draw five intermediate rainfall thresholds in the zone between the lower limit and the upper limit rainfall thresholds,

representing the 20 %, 30 %, 40 %, 50 % and 60 % probability of occurrence of landslide events (Table 3 and Fig. 8). Within this chart, any rainfall event exceeding the $PP_r$ x % threshold has the x % probability to generate a landslide event in the study area. Further probabilities were not possible to compute due to lack of data.

To analyse the performance of the $PP_r$ rainfall thresholds presented in Fig. 8, the ROC metrics were calculated and area summarized in Table 5. As expected, the False Alarm rate decreases as the $PP_r$ increases, and the same occurs with the True

positive rate and the False positive rate. According to the Threat Score (TS), the $PP_r$ 40 % and the $PP_r$ 50 % are the rainfall thresholds with the best performance (TS = 0.34 in both cases).

The return period of the rainfall associated to each calculated threshold presents a wide variation according to the considered number of consecutive days of accumulated rainfall (Fig. 9). As a rule, shorter durations (below 10 days) present a high return period in comparison with longer durations, independently on the type of rainfall threshold. In the cases of the upper

limit threshold, the $PP_r$ 60 % and the $PP_r$ 50 % thresholds, the obtained return periods for the shorter durations are less realistic and the corresponding rainfall values were never registered in the rainfall data series of the Lisboa-Geofísico rain gauge.

Figure 9 also shows that rainfall threshold is easier to reach for periods ranging from 15 to 45 consecutive days, namely for the regression threshold (linear), the lower limit threshold, and the $PP_r$ 20 %, $PP_r$ 30 % and $PP_r$ 40 % thresholds. For the

mentioned durations these thresholds will be exceeded by rainfall events with return period less than 10 years. However, for durations longer than 45 consecutive days, the return period of the corresponding rainfall denotes an increase trend for all the thresholds, although the return period remains lower, when compared with periods less than 10 consecutive days.

Data summarized in Fig. 8 and 9 can be combined to better characterize any rainfall threshold. Taken as example the $PP_r$ 60 % threshold, we can state that the highest yearly probability for this threshold to be exceeded is 5 % (20 year-return period)

associated to 30 to 60 consecutive days. The probability of landslide occurrence is 60 % given rainfall exceeding the threshold. Therefore, the maximum yearly combined probability of a landslide event associated with the $PP_r$ 60 % threshold is 3 %.



## 4.4 Regional performance of the lower limit threshold

Although the rainfall thresholds for landslide occurrence were defined taking in consideration the landslide events registered up to 10 km distance from the reference rain gauge of Lisboa-Geofísico, we admit that the obtained thresholds may be valid for distances larger than 10 km. In accordance, the performance of the lower limit threshold was evaluated for each buffer

zone represented in Fig. 4. The ratio between the FN and the total set of landslide events (TP + FN) for the different buffer zones is summarized in Table 6. As expected, the lowest ratio (0.167) corresponds to the buffer zone of 10-15 km. The ratio remains relatively stable within buffer zones up to 50 km distance from the rain gauge (ratio ranging from 0.2 to 0.297), and increases significantly in the buffer zone of 50-60 km (0.5). Therefore, we can conclude that thresholds identified for the Lisboa-Geofísico rain gauge can be applied with reasonable confidence for the area within 50 km distance.

## 5 Discussion

This work describes a comprehensive methodology to establish rainfall thresholds based on a reference rain gauge located in an urban area. Along the work a few methodological issues were highlighted, which are discussed in the following sections.

### 5.1 The concept of landslide event

The concept of landslide event is not straightforward as it has been applied in literature to describe both a landslide or a set

of landslides usually related with a specific triggering factor, such as an intense rainstorm (Crozier and Glade, 1999; Zêzere et al., 2014). When the landslide event is a single landslide, generally there is no problem to identify the date of the event that will be related with daily rainfall data for the rainfall threshold assessment. However, when several landslides are triggered over consecutive days in a study area, this may be a source of bias for the rainfall threshold definition. Usually, a date between the start and the end of the rainfall event is selected (e.g. Gullà et al., 2012; Gariano et al., 2015b), and

therefore, a unique combination of rainfall quantity-duration is calculated. The selection of the landslide event date is critical for this method as it can lead to the overestimation of the threshold, particularly if the end date of a long-lasting rainfall event is chosen. In these cases, the chosen quantity-duration rainfall may be not representative of the triggering conditions of landslides that occurred in the beginning of the event. To address this problem, in this work a landslide event was considered as an individual landslide or a set of landslides that occurred on a precise date (day). Therefore, in those cases where

different landslides occurred in consecutive days, each day was considered as a landslide event and the corresponding antecedent rainfall was used for the rainfall threshold assessment. In addition when the activity period of a landslide was reported as lasting during several days, the first day of the period was considered for the landslide event.

### 5.2 The use of one or several rain gauges to assess rainfall thresholds

Several benefits and drawbacks can be outlined regarding the use of a single rain gauge or multiple rain gauges to assess

rainfall thresholds for landslide initiation. The use of multiple rain gauges is a typical option to assess rainfall thresholds (e.g.



Caine, 1980; Gariano et al., 2015b; Peruccacci et al., 2017). The main advantage lies in the proximity of the rain gauge from the landslides, which provides a better relationship between rainfall and landslide triggering. However, the rainfall thresholds obtained in different rain gauges may be biased due to the different topographic and physiographic context characterizing each point of rainfall measuring. In these circumstances, the obtained rainfall thresholds will be biased by the differences regarding the rainfall regime of each location. Therefore, the merging and comparison of several rainfall datasets obtained in different places should be preceded by the normalization of rainfall data.

In addition, this type of analysis demands a high density of rain gauges network, which is only available for recent years. In Europe the number of stations increased after 1960 and had a peak between 1980 and 1990 (Haylock et al., 2008). In Portugal, a reliable rain gauge network only exists since 1980, when the mean distance between neighboring rain gauges was about 7.9 km (Belo-Pereira et al., 2011). Therefore, the use of multiple rain gauges to assess rainfall thresholds in the Lisbon region before 1980 should increase the threshold uncertainty due to the very low density of the available rain gauges. The restriction of the analysis to the period 1980 – 2010 was a possible alternative to overcome this limitation. However, this option was not considered because the number of landslide events would be drastically reduced (from 96 to 15 landslide events, Fig. 2), and the same would happen concerning the number of available rainfall events that did not generated landslides. The reduction of data representativeness would decrease the reliability of obtained rainfall thresholds.

Our landslide database covers a 145-year period (from 1865 to 2010) and we decided to analyse the complete period; thus, the selection of a single rain gauge was inevitable. The Lisboa-Geofísico rain gauge has uninterrupted rainfall measurements since 1864 and it is one of the few rain gauges in Portugal with long-term rainfall dataset. The comparable rain gauges (Évora, Porto, Guarda and Coimbra) are located more than 100 km distance from Lisbon. A long time series is an important condition to apply the return period and the Gumbel probability to the rainfall data. Moreover, this also provides a more reliable relation between the rainfall conditions and landslides, strengthening the reliability of the obtained thresholds. This is particularly important for a climate with a great variability, as the one existing in the study area.

## 5.3 Identification of landslide rainfall-triggered events

The uncertainty related with the triggering factor is particularly high when newspapers are the main source of information and, additionally, when a long time series is been analysed. As a rule, only newsworthy content are reported by newspapers, which certainly create bias in the landslide database. For instance, landslides that generated human damage or occurred in urban environment are usually highlighted, which increases the probability of landslides triggered by human action to be included in the database. On contrary, landslides triggered by rainfall that did not generate any social or economic damage were probably unreported by newspapers. In addition, the long time elapsed since the occurrence of some landslides inhibits the use of recent methods and techniques to confirm the rainfall triggering. For example, the confirmation of landslide events using aerial photo interpretation is only possible in Portugal for the period after 1947.

Using field-based landslide inventories in the Lisbon Region, Zêzere et al. (2015) considered as rainfall-triggered landslide event any date for which at least five individual landslides are known to have occurred on natural slopes. This criterion



reduces the possibility of inclusion of landslide triggered by human action. However, this criterion cannot be used in the present study, because landslides reported in newspapers are certainly a small sample of total triggered landslides. Therefore, any reported date, even those reporting a single landslide, should be admitted as a 'landslide event candidate'. Addressing the issue of the triggering factor, those landslide events associated to rainfall combinations (quantity-duration) with return

period below 3 years were rejected as rainfall-triggered. This criterion proved to be suitable to distinguish between rainfall events that triggered and did not trigger landslide events in the study area. However, further investigation should be made in this topic, namely in other study areas.

## 5.4 The spatial representativeness of a rain gauge data series

The discussion on the spatial representativeness of a rain gauge data series used to assess rainfall thresholds for landslide

activity is scarce in literature, which is surprisingly taking into consideration the large number of papers dealing with empirical rainfall thresholds published in recent years. In previous work using multiple rain gauges, the distance between the gauge and the landslides is the criterion used to select the rain gauge (e.g. Brunetti et al., 2010; Peruccacci et al., 2017), but the discussion on the topic is scarce and different distances have been proposed for the same region. For instance, for the Calabria region (Italy) Vennari et al. (2014) used 12 km as limit distance, whereas Gariano et al. (2015b) used 5 km.

To the best of our knowledge the spatial representativeness of a single rain gauge used to assess rainfall thresholds was never addressed before. In this work, we applied a method to compute the critical distance based on the ratio of non-rainfall triggered landslide events and rainfall-triggered landslide events tested along several buffer zones starting from the rain gauge at 5, 10, 20, 30, 40, 50 and 60 km distance. Our method takes in account both the source of landslide data (newspapers) and the location of the rain gauge in the urban area. We acknowledge that this method is valid in urban areas,

as is the case of the Lisbon region, and can be applied in other zones with similar context. However, the method cannot be directly applied in non-urban areas, which is a drawback.

In addition, an effort was made to evaluate the regional performance of the lower limit threshold, which was proved to be applied with reasonable results up to 50 km distance from the rain gauge. It should be pointed out, that the climatic and topographic features of the study area allow for the spatial enlargement of the threshold. The rainfall regime of the region is

spatially consistent and it is mainly influenced by the atmospheric general circulation and by the oceanic proximity, with the same weather types associated with high precipitation (Trigo and DaCamara, 2000; Ramos et al., 2014). In addition, the orographic effect on the rainfall distribution is low in the region, which enlarges the spatial representativeness of the reference rain gauge. However, the distance where the thresholds can be applied will be always connected with high levels of uncertainty associated to the rainfall discontinuity both in space and time. Therefore, the consideration of the lower limit

threshold up to 50 km should be used only where no other threshold is available.





## 6. Conclusion

The definition of rainfall thresholds for landslide initiation is typically characterized by uncertainty, which makes the use of probabilistic approach highly recommended (e.g. Frattini et al., 2009; Berti et al., 2012). In this study a comprehensive method to assess the rainfall thresholds was applied using a centenary database of landslides occurred in the Lisbon region, from 1865 to 2010, combined with a rainfall dataset collected at the Lisboa-Geofísico rain gauge, with uninterrupted daily measurements since 1864. The identification of the critical rainfall combinations responsible for preparing and triggering the landslide events were identified by selecting the pairs (rainfall quantity-duration) with the highest return period. Rainfall events that did not generate landslides were also selected and included in the analysis.

The use of a single rain gauge to assess rainfall thresholds implies the definition of the geographical area where the thresholds can be applied. In this study we demonstrated that 10 km is the optimal distance to compute the rainfall thresholds, although these may be spatially extended with enough confidence up to 50 km. These distances are controlled by the climatic and physiographic characteristics of the study area and should not be directly extrapolated to other study areas.

The zone between the lower limit and the upper limit thresholds (where landslide events may occur) was analyzed following a probabilistic approach, based on the Positive Predictive rate. Therefore, a range of probabilities of landslide event were established associated to five intermediate thresholds (20 %, 30 %, 40 %, 50 % and 60 %), which allow quantifying the uncertainty. Additionally, the performance of each threshold was assessed using ROC metrics. This approach can be used within landslide early warning systems as different alert levels can be associated to different probabilities of landslide occurrence.

The probability of exceedance of any rainfall event combined with the probability of landslide occurrence given rainfall exceeding the threshold was also calculated. This information can be more informative to the decision makers responsible for spatial planning, although additional information is needed regarding the landslide magnitude and the spatial distribution of future landslides.

The probabilistic approach used in this study is based on very long time series of landslide events and rainfall measurements, which are seldom available. This is a serious constrain to the application of the methodology to other study areas where long time series of landslide events and rainfall measurements are not available. In any case, the use of landslide inventories covering long time periods is crucial to obtain reliable thresholds valid at the regional scale.

### Acknowledgments

This work is financed by national funds through the FCT - Foundation for Science and Technology, I.P., in the framework of the project FORLAND – Disastrous floods and landslides in Portugal: driving forces and applications for land use planning (PTDC/ATP-GEO/1660/2014).





Teresa Vaz is a PhD fellow funded by FCT (SFRH/BD/74716/2010). Sérgio Cruz Oliveira is Post-Doc fellow funded by FCT (SFRH/BPD/85827/2012). The event data impacts information and newspapers was provided by DISASTER database and by Ivânia Quaresma, Pedro Santos, and Susana Pereira.

The authors are thankful to the IDL by the Lisboa-Geofísico rainfall data.

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





Table 1. ROC metrics (according to Staley et al., 2013).

| | Formulation | Optimal value |
|---|---|---|
| True positive rate (TP$_r$) | $TP_R = \dfrac{TP}{TP + FN}$ | 1 |
| False positive rate (FP$_r$) | $FP_R = \dfrac{FP}{FP + TN}$ | 0 |
| False Alarm rate (FA$_r$) | $FA_R = \dfrac{FP}{TP + FP}$ | 0 |
| Treat Score (TS) | $TS = \dfrac{TP}{TP + FN + FP}$ | 1 |



**Table 2.** Ratio of non-rainfall triggered landslide events / rainfall-triggered landslide events for different buffer distance to the reference rain gauge.

| Distance to the rain gauge (km) | Non-rainfall triggered landslide events (a) | Rainfall triggered landslide events (b) | Ratio (a/b) |
|---|---|---|---|
| 5 | 51 | 78 | 0.65 |
| 10 | 60 | 96 | 0.63 |
| 15 | 67 | 101 | 0.66 |
| 20 | 69 | 105 | 0.66 |
| 30 | 78 | 117 | 0.67 |
| 40 | 86 | 125 | 0.69 |
| 50 | 88 | 128 | 0.69 |
| 60 | 92 | 131 | 0.70 |





**Table 3.** Equations of rainfall thresholds for landslide events in the Lisbon region.

|  |  | Equations |
| --- | --- | --- |
| Regression threshold (linear) |  | $y = 5.5D + 124.6$ |
| Regression threshold (potential) |  | $y = 67.8D^{0.46}$ |
| Lower limit threshold |  | $y = 4.4D + 56.5$ |
| Upper limit threshold |  | $y = 7.3D + 235.8$ |
| Intermediate thresholds | $PP_r$ 20 % | $y = 4.6D + 68.0$ |
|  | $PP_r$ 30 % | $y = 4.8D + 84.8$ |
|  | $PP_r$ 40 % | $y = 5.1D + 98.2$ |
|  | $PP_r$ 50 % | $y = 5.3D + 113.0$ |
|  | $PP_r$ 60 % | $y = 6.2D + 164.1$ |



**Table 4.** ROC metrics associated to the rainfall thresholds for landslide events in the Lisbon region.

| | Regression threshold (linear) | Regression threshold (potential) | Lower limit threshold | Upper limit threshold |
|---|---|---|---|---|
| True Positive (TP) | 38 | 39 | 96 | 3 |
| False Negative (FN) | 58 | 57 | 0 | 93 |
| False Positive (FP) | 34 | 48 | 527 | 0 |
| True Negative (TN) | 1394 | 1380 | 901 | 1428 |
| True Positive rate (TP$_r$) | 0.40 | 0.41 | 1 | 0.03 |
| False Positive rate (FP$_r$) | 0.02 | 0.03 | 0.37 | 0 |
| False Alarm rate (FA$_r$) | 0.47 | 0.55 | 0.85 | 0 |
| Threat Score (TS) | 0.29 | 0.27 | 0.15 | 0.03 |
| Positive predictive rate (PP$_r$) | 0.53 | 0.45 | 0.15 | 1 |



**Table 5.** ROC metrics associated to intermediate thresholds based on the probability of landslide events in the zone between the lower limit threshold and the upper limit threshold.

|  | PP$_r$ 20 % | PP$_r$ 30 % | PP$_r$ 40 % | PP$_r$ 50 % | PP$_r$ 60 % |
|---|---|---|---|---|---|
| True Positive (TP) | 91 | 76 | 66 | 50 | 15 |
| False Negative (FN) | 5 | 20 | 30 | 46 | 81 |
| False Positive (FP) | 364 | 177 | 99 | 50 | 10 |
| True Negative (TN) | 1064 | 1251 | 1329 | 1378 | 1418 |
| True Positive rate (TP$_r$) | 0.95 | 0.79 | 0.69 | 0.52 | 0.16 |
| False Positive rate (FP$_r$) | 0.25 | 0.12 | 0.07 | 0.04 | 0.01 |
| False Alarm rate (FA$_r$) | 0.80 | 0.70 | 0.60 | 0.50 | 0.40 |
| Threat Score (TS) | 0.20 | 0.28 | 0.34 | 0.34 | 0.14 |
| Positive Predictive rate (PP$_r$) | 0.20 | 0.30 | 0.40 | 0.50 | 0.60 |





**Table 6.** Ratio FN/(TP+FN) considering the lower limit rainfall threshold for different buffer distances to the rain gauge.

| Distance (km) | True Positive (TP) | False Negative (FN) | Ratio FN/(TP+FN) |
|---|---|---|---|
| [10-15[ | 20 | 4 | 0.167 |
| [15-20[ | 7 | 2 | 0.222 |
| [20-30[ | 26 | 11 | 0.297 |
| [30-40[ | 23 | 7 | 0.233 |
| [40-50[ | 8 | 2 | 0.200 |
| [50-60[ | 5 | 5 | 0.500 |





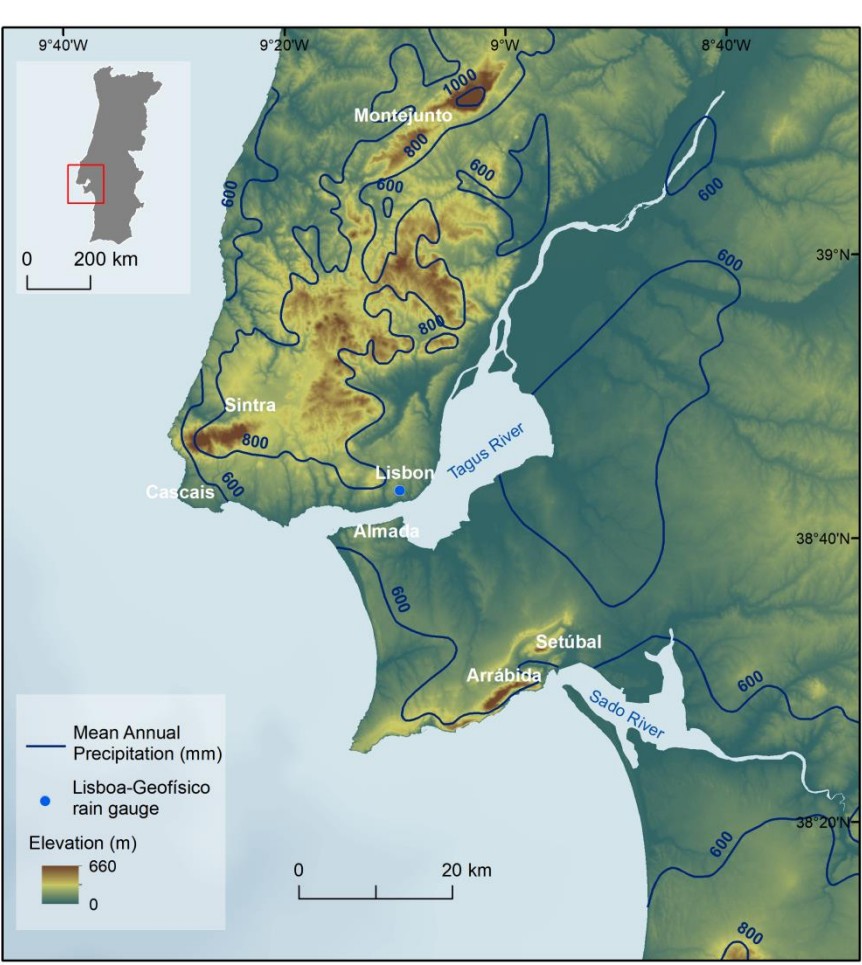

5    **Figure 1.** Elevation and mean annual precipitation in the study area (source: Daveau et al., 1977).





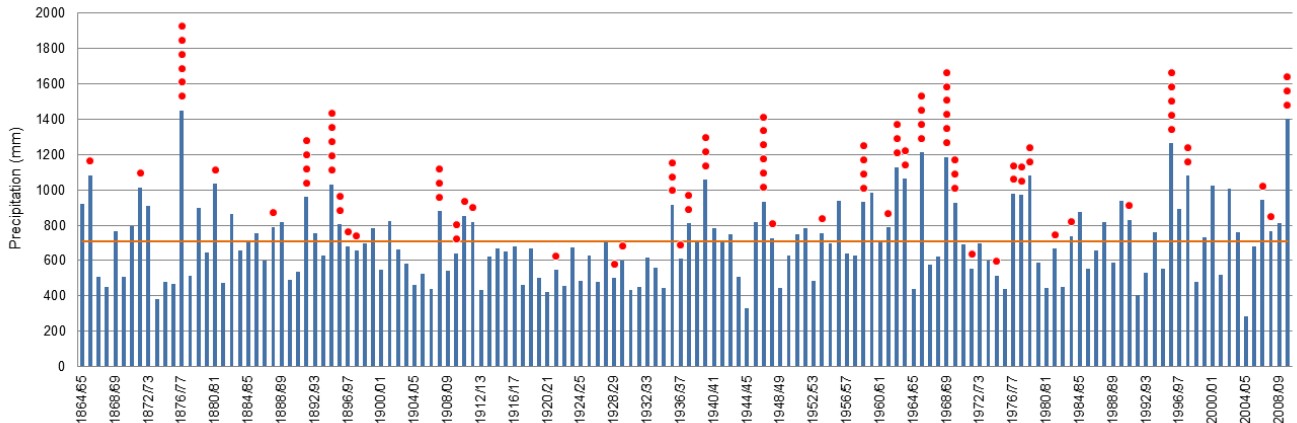

**Figure 2.** Annual precipitation (climatological year: September to August) at Lisboa-Geofísico rain gauge for the period 1864/65 - 2009/10. Orange line symbolizes the Mean Annual Precipitation (MAP); and red dots represent landslide events.





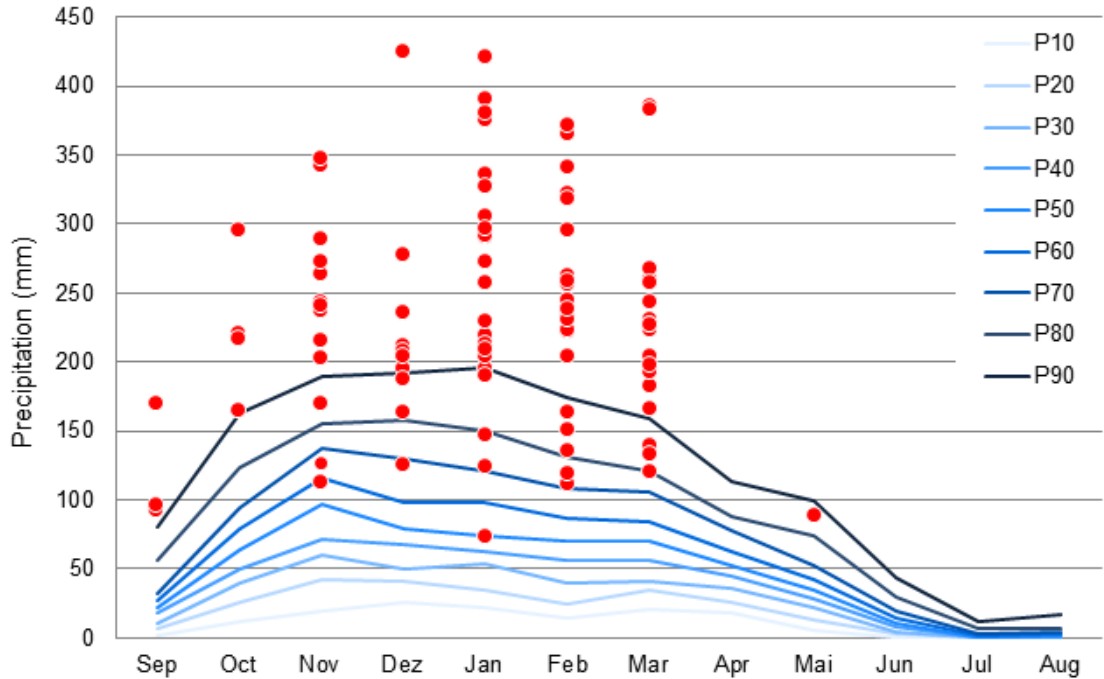

**Figure 3.** Monthly precipitation percentiles at Lisboa-Geofísico rain gauge for the period 1864/65 - 2009/10. Red dots symbolize the 30-day cumulative absolute antecedent rainfall for each landslides event.





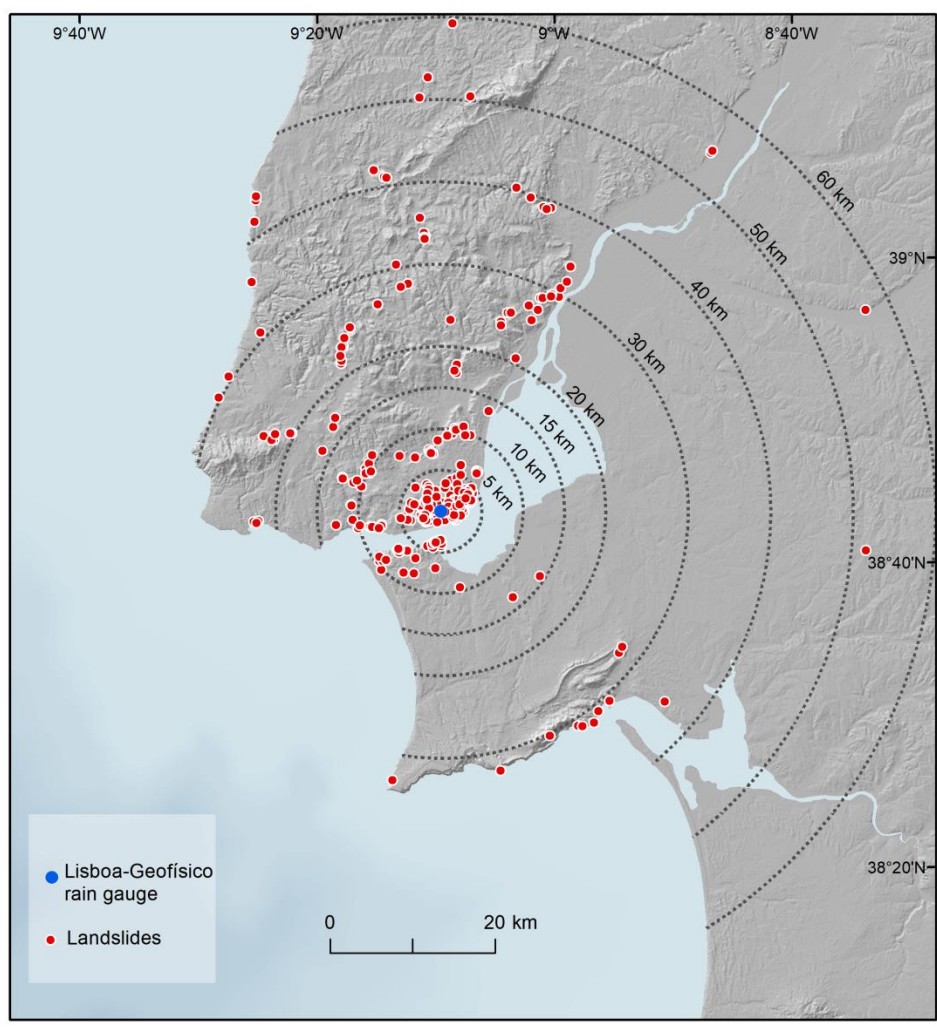

**Figure 4.** Distribution of landslides in the Lisbon region (1865/2010) and buffers plotted from the reference rain gauge.





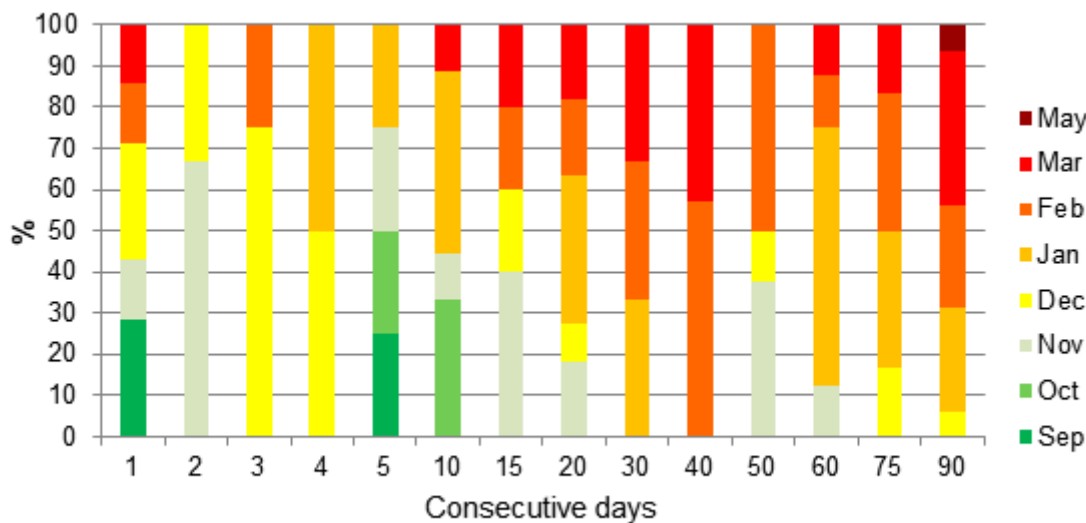

**Figure 5.** Monthly frequency of the rainfall-triggered landslide events according to the duration of the rainfall period.




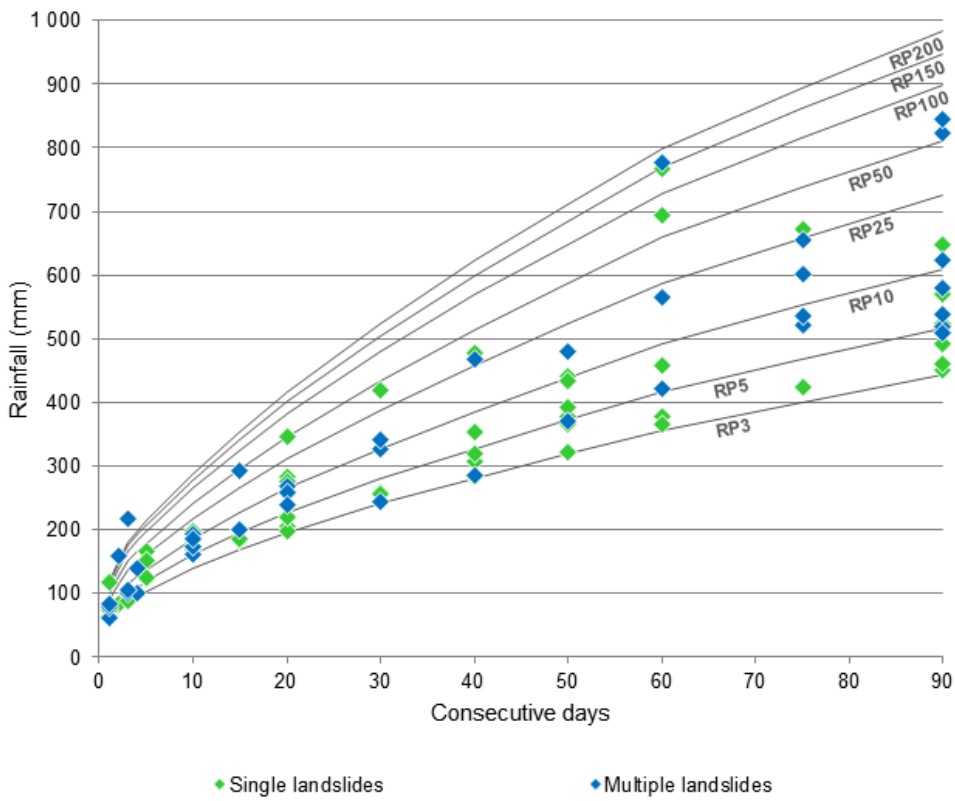

**Figure 6.** Critical rainfall combination quantity–duration that resulted in landslide events (single or multiple landslides) and Return Period (RP) for 3, 5, 10, 25, 50, 100, 150 and 200 years. Distance up to 10 km from the reference rain gauge.





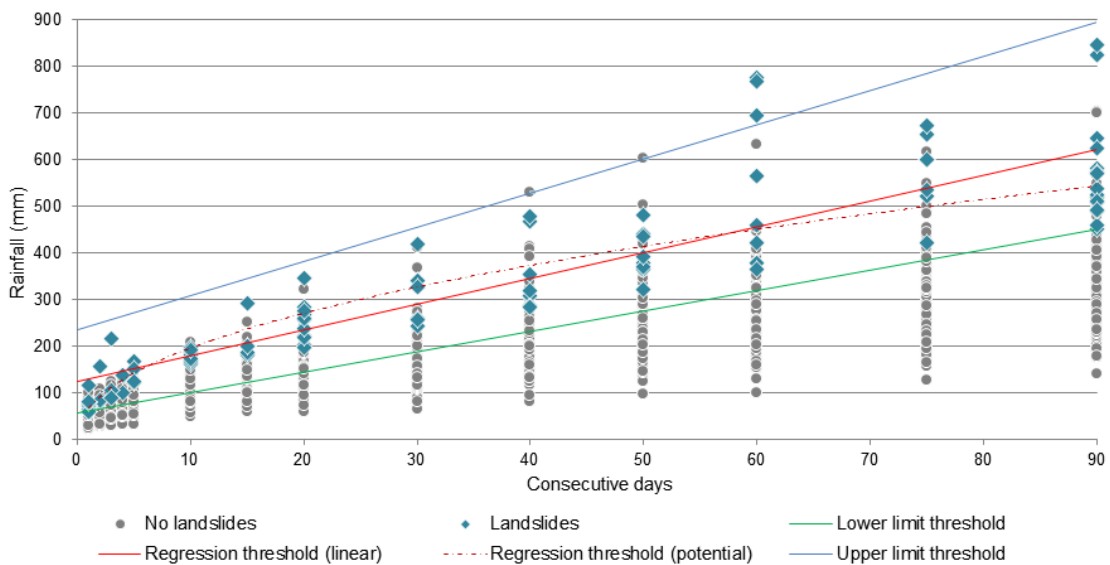

**Figure 7.** Rainfall quantity–duration thresholds for landslide events in the Lisbon region (1865 to 2010). Distance up to 10 km from the reference rain gauge.





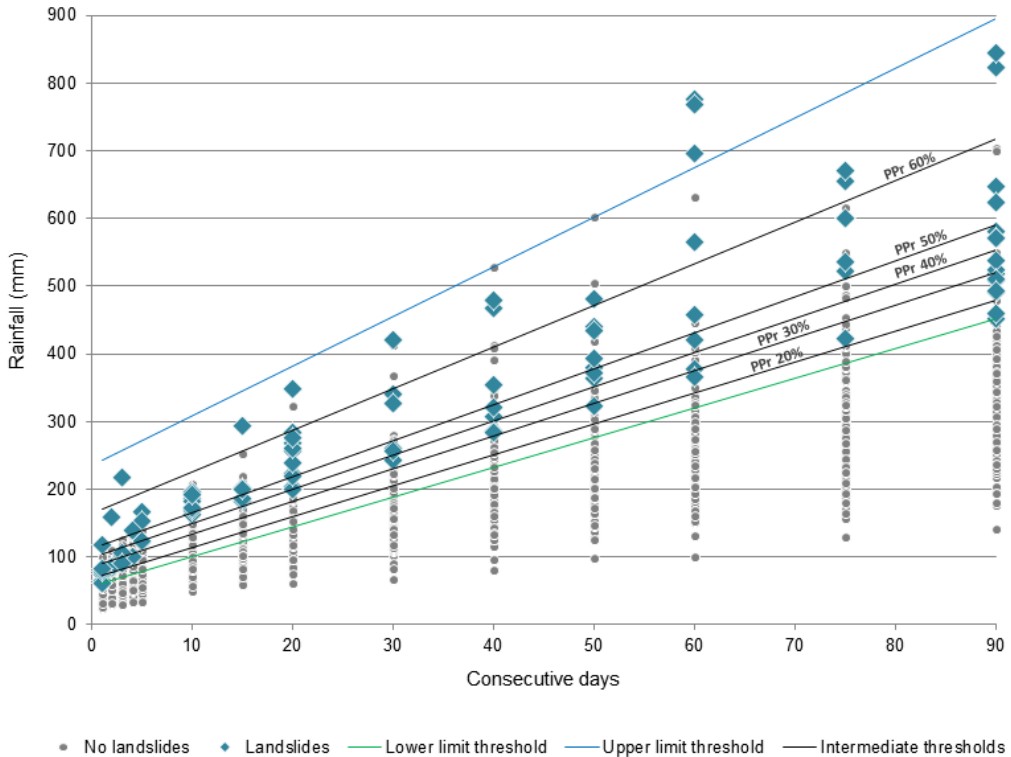

**Figure 8.** Intermediate thresholds based on the probability of landslide events (20 %, 30 %, 40 %, 50 % and 60 %) in the zone between the lower limit threshold and the upper limit threshold.



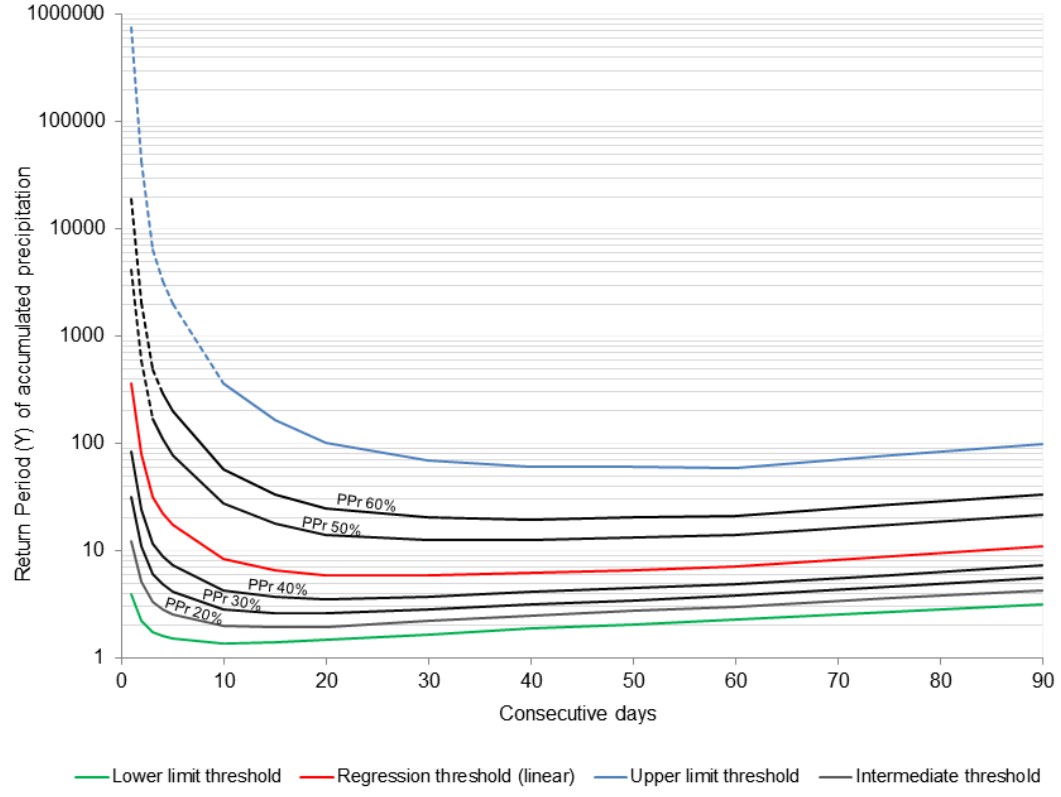

**Figure 9.** Return period of rainfall thresholds computed for the Lisbon region (linear regression, lower limit, upper limit and intermediate thresholds PP$_r$ 20 %, PP$_r$ 30 %, PP$_r$ 40 %, PP$_r$ 50 %, and PP$_r$ 60 %). Dashed lines represent conditions never registered in the rain gauge.

