# Peer review of "Regional rainfall thresholds for landslide occurrence using a centenary database"

_Natural Hazards and Earth System Sciences, 2017_

## Referee Comment (RC1) · Anonymous Referee #1 · 10 Nov 2017

The contribution "Regional rainfall thresholds for landslide occurrence using a centenary database" by Teresa Vaz and co-authors presents a procedure for the definition and the evaluation of rainfall thresholds for landslide occurrence, based on a huge database of landslide occurrence and a long series of daily rainfall data gathered from one rain gauge.

The paper is clear, well written and potentially publishable. It follows the IMRaD structure, with a fluent language; as a result, the manuscript is easily readable. Tables and figures are useful for a better understanding of method and results. Actually, the proposed approach is not very new but the use of such a big database is not common in the literature. Moreover, the analysis of the spatial representativeness of the rain gauge data series is well structured and brings novelty and interest to the paper. The use of

a single rain gauge can be considered a drawback of the work, but I understand that with such a long series of data, it was the only way to present the proposed procedure.

My opinion is that the paper deserves to be published in NHESS journal after some revisions. I have some comments and suggestions that should be addressed before the manuscript can be accepted for publication.

I have a main comment regarding the method proposed for defining the thresholds. I partially disagree with the point stated at page 6, line 11, where the Authors say: "The pair with the highest return period was considered as the critical rainfall combination responsible for triggering the landslide events". This can be true, but not always, also from a physical point of view. Indeed, this can be seen also by inspecting Figure 6. In wide the literature concerning landslides and thresholds, there are several cases in which landslides are triggered by (or are associated with) rainfall events without considering their return periods. In addition, in some cases, the rainfall events associated to the landslide triggering are not very high (e.g., in terms of cumulated rainfall). Moreover, considering the rainfall return period to define the triggering rainfall events imply a somehow "linear" relationship between rainfall and landslide, and we know that this is not exactly true. On the other hand, I acknowledge that in any case and for any proposed method, a simplification is needed. Further on in the text (lines 16-18 and also in the discussion at page 14, lines 3-7), the Authors state that they used only the landslide events with critical rainfall combination having return period > 3 years. This point seems questionable too. Why do not use only landslides for which rainfall triggering is known? Even because the 3-year boundary is arbitrary, as acknowledged by the Authors. Moreover, I cannot understand how this selection may limit the possibility to exclude landslides not triggered by rainfall (e.g. human action). I think that a rainfall event with a return period lower than 3 years can induce a landslide, given that this is a process controlled not only by the triggering rainfall but also by other variables, like e.g. the antecedent rainfall, or the antecedent soil moisture conditions, and so on. As an example, a rainfall characterized by a (cumulated rainfall-duration) combination with

a low return period can trigger a landslide if the antecedent conditions are worse (negative) for slope stability. Actually, this is my point of view, and I do not want to criticize or reject the method proposed in this work, but I want to stimulate a discussion about this point. Thus, after this long comment, I would ask the Authors to better justify this part of the procedure in order to make it more shareable.

Furthermore, in several parts of the text, the Authors state that they performed a calibration and a validation of the thresholds, by using – briefly resuming – years with landslides and years without landslides. In my opinion, this is actually only a calibration or, at most, an optimization, given that the same temporal rainfall and landslide datasets are used for constructing the contingency matrix. An independent validation could be performed by splitting the database in two non-overlapping periods and testing the performance of the thresholds (calibrated using the first subset) by verifying the number of TP, FN, FP, TN, and all the indexes considering the second subset, as done e.g., by Martelloni et al. (2012), Giannecchini et al. (2012), Segoni et al. (2014), Gariano et al. (2015), Piciullo et al. (2017). Thus, I suggest to better clarify this issue whether performing an independent validation or rephrasing the sentences by using calibration/optimization/optimal-definition/objective-definition instead of validation.

In Section 3.3, the method for drawing the lower and upper limit thresholds is not specified. I see the reference to the work of Glade and co-authors, but I would ask to add more details (e.g. how many pairs were used to draw the thresholds, how they were selected, . . .).

A section or a sub-section with the description of the landslide database is missing (even if some references are provided). However, I suggest adding a subsection (of sections 2 or 3) with a brief description of the landslide database. This subsection could contain, among the others, some information about the percentage of known/unknown types of the landslides, about the precision in the location of the landslides, about the temporal accuracy of the records, and so on.

In Section 4.1, a description and a comment of the values reported in Table 2 is provided. Actually, the differences of the Ratio a/b seems very little: only in the second decimal place. This is not reported in the text, and I think that should be acknowledged. The same is for what reported in Section 4.2 (Page 10, Line 31) where Authors state that "regression thresholds, linear or potential can be used as acceptable thresholds to predict landslide events...". Actually, this can be considered true, but only the index FAr returns very good values (as can be noted in Table 4). This should be acknowledged by the Authors too.

At the end of section 4.4, Authors state that "thresholds identified for the Lisboa-Geofisico rain gauge can be applied with reasonable confidence for an area within 50 km distance". This sentence is very difficult to justify, and I think that the ratio a/b or the ratio FN/(TP+FN) cannot be used to justify that assumption. In my opinion, the number of TP and FN are related to the efficiency of prediction model rather than to the representativeness of the rain gauge. I suggest to better justify that sentence or to modify it.

In the discussion, a section/paragraph with a brief discussion about the possible application of the thresholds in an operational or prototypal landslide early warning system could introduce more appeal to the work, also in the perspective of the topic of the Special Issue. I list some works that could be mentioned at this regards: Aleotti (2004); Tiranti and Rabuffetti (2010); Segoni et al. (2015); Calvello et al. (2015); Piciullo et al. (2017).

I have also some comments about some words used in the text.

First, I would suggest using the word "method" (i.e., a systematic way of doing something, implying an orderly logical arrangement, usually in steps) instead of the word "methodology" (i.e., a system of methods followed in a particular discipline). Likewise, I would suggest using "type" (a subdivision of a particular kind of thing) instead of "typology" (a classification according to general type).

Moreover, according to the words "precipitation" and "rainfall", I would ask if the Authors know the type of data recorded by the Lisbon rain gauge. If it includes all types of precipitation (i.e., also snow) it is fair to use the term "precipitation"; otherwise I would suggest using simply "rainfall".

Finally, I am somewhat dubious about using the word "quantity", as for "rainfall quantity". I would suggest the most used "cumulated rainfall".

Moreover, some specific comments are listed below.

Page 2, Line 4: I suggest to add "and variables" after "rainfall measurements".

Page 2, Line 10: I suggest to change "triggered during periods of rainfall of short durations" into "triggered by short and intense rainfall".

Page 2, Line 31: Nikolopoulos et al. (2014; 2015) addressed the issue of rain gauge representativeness.

Page 4, Lines 3-5: this sentence is quite complicate and hard to follow. Please reword.

Page 5, Line 16: the equation is quite ambiguous. Please clarify it and add an example.

Page 9, Lines 19-20: I suggest to rephrase the sentence "The relationship between critical duration and month..." as "The monthly distribution of critical durations". It seems clearer to me.

Page 10, Line 3: the sentence "however none of them ensure the false negatives occurrence" is not clear. Do the Authors mean "none of them ensure a low number of FN"?

Page 11, Line 2: does the PPr index indicate probability of landslide occurrence or probability of landslide prediction?

Page 14, Lines 11-13: in the cited works, the criteria used to select the representative rain gauges were based not only on the topographic distance between gauge and

landslide, but also to elevation difference and morphological settings.

Tables

Tables 3, 4, and 5 could be merged into one figure (by transposing tables 4 and 5 and by merging them to table 3). They have the same number of rows and columns. Moreover, I suggest using only the acronyms (TP, FN, FP, TN, TPr, FPr, FAr, TS, PPr) instead of the entire names of these parameters and indexes.

In Table 3, I suggest to use a specific letter for indicating the variable of the rainfall in the threshold equation, e.g. "R" instead of "y".

Figures

Figure 2.

I would suggest adding a moving average, other than the MAP line.

The red dots represent only the landslides in the 10 km buffer?

Figure 3.

In order to avoid misunderstanding, I suggest using another shape or another colour to symbolize the 30-day cumulative antecedent rainfall, instead of the red dots (used in figure 2 to represent the landslides).

Please correct "Dez" and "Mai" in the labels of x-axes.

Some technical corrections:

Page 3, Line 20: delete the space after the open bracket.

Page 5, Line 16: add a "+" before "..." in the equation.

Page 9, Line 14: delete "the" before "Fig. 3".

Page 9, Line 16: correct "70 percentile" and "90 percentile". I suggest to use "70th" or "70%".

Page 9, Line 27: delete "the" before "Fig. 6".

Page 9 and followings (also in Table 3): I suggest to use a specific letter for indicating the variable of the rainfall in the threshold equation, e.g. "R" instead of "y".

References (some of them are already cited in the manuscript)

Aleotti, P.: A warning system for rainfall-induced shallow failures, Eng. Geol., 73, 247–265, doi:10.1016/j.enggeo.2004.01.007, 2004.

Calvello, M., d'Orsi, R.N., Piciullo, L., Paes, N., Magalhaes, M.A., Lacerda, W.A.: The Rio de Janeiro early warning system for rainfall-induced landslides: analysis of performance for the years 2010–2013, Int. J. Disast. Risk Reduc., 12, 3–15, doi:10.1016/j.ijdrr.2014.10.005, 2015.

Gariano, S.L., Brunetti, M.T., Iovine, G., Melillo, M., Peruccacci, S., Terranova, O.G., Vennari, C., Guzzetti, F.: Calibration and validation of rainfall thresholds for shallow landslide forecasting in Sicily, southern Italy, Geomorphology, 228, 653–665, doi:10.1007/s11069-014-1129-0, 2015.

Giannecchini, R., Galanti, Y., D'Amato Avanzi, G.: Critical rainfall thresholds for triggering shallow landslides in the Serchio River Valley (Tuscany, Italy), Nat. Hazards Earth Syst. Sci., 12, 829–842, doi:10.5194/nhess-12-829-2012, 2012.

Martelloni, G., Segoni, S., Fanti, R., Catani, F.: Rainfall thresholds for the forecasting of landslide occurrence at regional scale, Landslides, 9, 485–495, doi:10.1007/s10346-011-0308-2, 2012.

Nikolopoulos, E.I., Borga, M., Creutin, J.D., Marra, F.: Estimation of debris flow triggering rainfall: Influence of rain gauge density and interpolation methods, Geomorphology, 243, 40–50, doi:10.1016/j.geomorph.2015.04.028, 2015.

Nikolopoulos, E.I., Crema, S., Marchi, L., Marra, F., Guzzetti, F., Borga, M.: Impact of uncertainty in rainfall estimation on the identification of rainfall thresholds for debris flow

occurrence, Geomorphology, 221, 286-297, doi:10.1016/j.geomorph.2014.06.015, 2014.

Piciullo, L., Gariano, S.L., Melillo, M., Brunetti, M.T., Peruccacci, S., Guzzetti, F., Calvello, M.: Definition and performance of a threshold-based regional early warning model for rainfall-induced landslides, Landslides, 14, 995-1008, doi: 10.1007/s10346-016-0750-2, 2017.

Segoni, S., Rossi, G., Rosi, A., Catani, F.: Landslides triggered by rainfall: a semiauto-mated procedure to define consistent intensity-duration thresholds, Comput. Geosci., 63, 123–131, doi:10.1016/j.cageo.2013.10.009, 2014.

Segoni, S., Rosi, A., Rossi, G., Catani, F., Casagli, N.: Analysing the relationship between rainfalls and landslides to define a mosaic of triggering thresholds for re-gional scale warning systems, Nat. Hazards Earth Syst. Sci., 14, 2637–2648, doi:10.5194/nhess-14-2637-2014, 2014.

Segoni, S., Battistini, A., Rossi, G., Rosi, A., Lagomarsino, D., Catani, F., Moretti, S., Casagli, N.: Technical note: an operational landslide early warning system at regional scale based on space–time variable rainfall thresholds. Nat. Hazards Earth Syst. Sci., 15, 853–861, doi:10.5194/nhess-15-853-2015, 2015.

Tiranti, D., Rabuffetti, D.: Estimation of rainfall thresholds triggering shallow implemen-tation, Landslides, 7, 471–481, doi:10.1007/s10346-010-0198-8, 2010.

---

## Referee Comment (RC2) · Anonymous Referee #2 · 22 Nov 2017

The paper by Vaz et al. deals with the identification of landslides triggering thresholds. The paper is well written, even if the topic is highly studied and some drawbacks are found. The most critical issue is due to the fact that landslides are not classified in terms of mechanism, material or volume (3.1 paragraph, line 23). I don't know if the authors could provide at least one these, but I think that an effort in at least one of them could affect the final results. In fact, antecedent rainfalls are very important for landslides affecting soils, but they could be less relevant for rockfalls... Most of the adopted landslides are located in urban area, where the fall of walls and some artificial cut could have been termed landslides. Several other landslides are located close to the sea, affecting the sea cliff, are the authors sure that they were not triggered by waves? I understand the attempt to find a filtering criterion in Page 6 lines 15 -19, but

as the authors say it is arbitrary and it is in contrast with the analysis they perform within 10 km, where most of them could be anthropic-induced (pag. 8 lines 23-25). Maybe, thresholds could be evaluated in the range 5 - 10 km, excluding urban landslides. All the performed analyses and the considerations carried out are reasonable and well-described, but main drawbacks are the input data.

minor issues: pag 2 line 8 change depth with height pag 3 line 6 change quantity with quantify pag 4 line 3 remove brackets for April etc pag 4 line 24 change along with In pag 9 line 13 most rainy pag 9 line 27 change detaches with identifies Fig 1 it is better to categorize the elevation Fig 3 some labels are wrong , i.e. Dez

---

## Author Comment (AC1) · 26 Jan 2018

Hazards Earth Syst. Sci. Discuss., https://doi.org/10.5194/nhess-2017-362
Regional rainfall thresholds for landslide occurrence using a centenary database, by Teresa Vaz, José Luís Zêzere, Susana Pereira, Sérgio C. Oliveira, Ricardo A. C. Garcia, and Ivânia Quaresma

**Reply to Anonymous Referee #1**

The contribution "Regional rainfall thresholds for landslide occurrence using a centenary database" by Teresa Vaz and co-authors presents a procedure for the definition and the evaluation of rainfall thresholds for landslide occurrence, based on a huge database of landslide occurrence and a long series of daily rainfall data gathered from one rain gauge.
The paper is clear, well written and potentially publishable. It follows the IMRaD structure, with a fluent language; as a result, the manuscript is easily readable. Tables and figures are useful for a better understanding of method and results. Actually, the proposed approach is not very new but the use of such a big database is not common in the literature. Moreover, the analysis of the spatial representativeness of the rain gauge data series is well structured and brings novelty and interest to the paper. The use of a single rain gauge can be considered a drawback of the work, but I understand that with such a long series of data, it was the only way to present the proposed procedure.
My opinion is that the paper deserves to be published in NHESS journal after some revisions. I have some comments and suggestions that should be addressed before the manuscript can be accepted for publication.

We appreciate the comments given by the reviewer #1. The paper will be changed according to his/her pertinent suggestions. Detailed answers to the reviewer comments are presented next, item by item. The reviewer comments are presented in black, followed by our answer in blue.

1. I have a main comment regarding the method proposed for defining the thresholds. I partially disagree with the point stated at page 6, line 11, where the Authors say: "The pair with the highest return period was considered as the critical rainfall combination responsible for triggering the landslide events". This can be true, but not always, also from a physical point of view. Indeed, this can be seen also by inspecting Figure 6. In wide the literature concerning landslides and thresholds, there are several cases in which landslides are triggered by (or are associated with) rainfall events without considering their return periods. In addition, in some cases, the rainfall events associated to the landslide triggering are not very high (e.g., in terms of cumulated rainfall). Moreover, considering the rainfall return period to define the triggering rainfall events imply a somehow "linear" relationship between rainfall and landslide, and we know that this is not exactly true. On the other hand, I acknowledge that in any case and for any proposed method, a simplification is needed.

We acknowledge the reviewer comments. To answer this topic a new section will be included in the discussion as follows:

"5.3 Empirical definition of critical rainfall period

Identifying the rainfall responsible for the landslide occurrence is the basis for any empirical rainfall threshold calculation. A range of procedures to define the rainfall critical period associated to landslide events have been proposed in literature (e.g. Guzzetti et al.;2007;

Segoni et al., 2014). Moreover, even the definition of critical rainfall is not straightforward. Aleotti (2004) defined the critical rainfall as the rainfall period starting when a shark increase in rainfall intensity is identified and ending when the first landslide is triggered. Therefore, in such circumstances, the cumulative rainfall before the rainfall increase is considered as antecedent rainfall and is not included in the critical rainfall. Brunetti et al. (2010) and Peruccacci et al. (2012, 2017) use the concept of "rainfall event", as a period of continuous rainfall separated by a dry period, with a seasonal variability concerning the length of the dry period (48 h in the dry season and 96 h in the wet season). In our study, the critical rainfall joints together the antecedent rainfall (acting as a landslide preparatory factor) and the rainfall that triggered the landslide event. Our procedure to define the critical rainfall combination, responsible for preparing/triggering the landslide event is based on the return period calculation, by selecting the cumulative rainfall with the highest return period.

This approach has the advantage of being an objective method easily reproducible to other areas and provides rainfall thresholds with the most optimistic results concerning the ROC metrics. However, the use of the return period implies a 'rigid' statistical relationship between the rainfall and landslides, which does not always occur. Moreover, in some cases, the cumulated rainfall associated to the landslide triggering is not very high. However, the use of other empirical procedure, as the previous mentioned, to define the critical rainfall period are based on subjective decisions, like the duration of the dry period to bound the rainfall events. Probably, the identification of the critical rainfall period for a landslide event is only possible to determine with precision using coupled geotechnical and transient hydrological physical models. However, each slope is a unique system and the rainfall is not uniform both in time and space, which explain the difficulty to establish rainfall thresholds based on physically-based models at the regional scale."

2. Further on in the text (lines 16-18 and also in the discussion at page 14, lines 3-7), the Authors state that they used only the landslide events with critical rainfall combination having return period > 3 years. This point seems questionable too. Why do not use only landslides for which rainfall triggering is known? Even because the 3-year boundary is arbitrary, as acknowledged by the Authors. Moreover, I cannot understand how this selection may limit the possibility to exclude landslides not triggered by rainfall (e.g. human action). I think that a rainfall event with a return period lower than 3 years can induce a landslide, given that this is a process controlled not only by the triggering rainfall but also by other variables, like e.g. the antecedent rainfall, or the antecedent soil moisture conditions, and so on. As an example, a rainfall characterized by a (cumulated rainfall-duration) combination with a low return period can trigger a landslide if the antecedent conditions are worse (negative) for slope stability. Actually, this is my point of view, and I do not want to criticize or reject the method proposed in this work, but I want to stimulate a discussion about this point. Thus, after this long comment, I would ask the Authors to better justify this part of the procedure in order to make it more shareable.

We acknowledge the reviewer comments. Firstly, it should be pointed out, that fall of walls and instabilities directly resulting from engineering works were rejected. However, the historical reports in newspapers are not enough clear in several cases to conclude about the rainfall triggering of landslides, especially using a centenary landslide database like this one.
We acknowledge the importance of the antecedent rainfall for landslide events, as stated by the reviewer, but the antecedent rainfall is considered together with the triggering rainfall in our thresholds. We believe this will be enough clear in the new version of the manuscript.
Addressing the issue of the triggering factor, we considered the criterion of return period above 3 years suitable to distinguish between rainfall events that triggered and did not trigger landslide events in the study area. Using field-based landslide inventories in the Lisbon Region, Zêzere et al. (2015) showed that only 12% of landslide events triggered by rainfall have cumulated rainfall return period below 3 years and landslide events were not registered with

rainfall conditions with return period below 2 years. Given our data source feature (based in newspaper) and our study area (integrated in an urban area) a more conservative boundary was preferred, and therefore, the selection of 3 years-return period.

This criterion can eventually eliminate some (few) landslide events triggered by rainfall in the study area. However, the possibility to include non-rainfall triggered landslide events would increase, not applying this criterion. The inclusion of non-rainfall triggered landslides in the analysis would bias the rainfall thresholds as well as the ROC metrics, generating a higher number of undesirable false alarms. We will discuss this topic in the section 5.4. In addition, the non-rainfall triggered landslide events will be plotted in figures 2,3,6 and 7.

3. Furthermore, in several parts of the text, the Authors state that they performed a calibration and a validation of the thresholds, by using – briefly resuming – years with landslides and years without landslides. In my opinion, this is actually only a calibration or, at most, an optimization, given that the same temporal rainfall and landslide datasets are used for constructing the contingency matrix. An independent validation could be performed by splitting the database in two non-overlapping periods and testing the performance of the thresholds (calibrated using the first subset) by verifying the number of TP, FN, FP, TN, and all the indexes considering the second subset, as done e.g., by Martelloni et al. (2012), Giannecchini et al. (2012), Segoni et al. (2014), Gariano et al. (2015), Piciullo et al. (2017). Thus, I suggest to better clarify this issue whether performing an independent validation or rephrasing the sentences by using calibration/optimization/optimal-definition/objective-definition instead of validation.

We agree with the comment of the reviewer concerning the independent validation of the thresholds. In accordance, we will use the word "calibration" instead of "validation". However, we would like to thank the reviewer for the suggestion and we will perform a validation of the thresholds in a future work.

4. In Section 3.3, the method for drawing the lower and upper limit thresholds is not specified. I see the reference to the work of Glade and co-authors, but I would ask to add more details (e.g. how many pairs were used to draw the thresholds, how they were selected, : : :).

The lower limit and the upper limit rainfall thresholds were defined by linear regression both based on two pairs. The lower limit was established by selecting iteratively two landslide events associated to different durations with the lowest values of cumulated critical rainfall and ensuring that the complete set of landslide events fall above the threshold.

The upper limit was established by selecting iteratively two highest pairs (cumulated rainfall/duration) that did not triggered landslides and ensuring that the complete set of non-landslide events fall below the threshold.

A new piece of text will be included in the new version of the manuscript in order to add more details to the used method.

5. A section or a sub-section with the description of the landslide database is missing (even if some references are provided). However, I suggest adding a subsection (of sections 2 or 3) with a brief description of the landslide database. This subsection could contain, among the others, some information about the percentage of known/unknown types of the landslides, about the precision in the location of the landslides, about the temporal accuracy of the records, and so on.

We acknowledge the comment of the reviewer. In accordance, a description addressing this topic will be included. Likewise, the subsection 3.1 will be divided in two new subsections: "3.1

Identification of landslide events" and "3.2 Selection of rain gauge and identification of critical rainfall combinations".
The new subsection 3.1. will include the following paragraph:

"Only landslides with at least 1 day of accuracy were included in the database. The spatial accuracy of landslides cases was classified, following Zêzere et al. (2014), in 5 classes: (i) location with exact coordinates (accuracy associated with scale 1:1 000); (ii) location based on local toponymy (accuracy associated with scale 1:10 000); (iii) location based on local geomorphology (accuracy associated with scale 1: 25 000 scale); (iv) location in the centroid of the parish; and (v) location in the centroid of the council. A total of 400 landslide cases were inventoried being the majority (83%) located with accuracy corresponding to classes (i) to (iii). These landslides affected clay (40.24%), sandstone and conglomerate (22.52%), limestone (16.52%), volcanic (11.11%), marly and marly limestone (9.01%) and granite (0.60%). The landslide type was classified following the Cruden and Varnes (1996) classification scheme. The slides are the dominant landslide type in the database (53.8%), followed by falls (14.4%). Flows and complex slope movements are less representative (2.4% and 1.5%, respectively). The landslide type is frequently unknown (27.9%)."

6. In Section 4.1, a description and a comment of the values reported in Table 2 is provided. Actually, the differences of the Ratio a/b seems very little: only in the second decimal place. This is not reported in the text, and I think that should be acknowledged.

   This information will be added in the new version of the manuscript.

7. The same is for what reported in Section 4.2 (Page 10, Line 31) where Authors state that "regression thresholds, linear or potential can be used as acceptable thresholds to predict landslide events…". Actually, this can be considered true, but only the index Far returns very good values (as can be noted in Table 4). This should be acknowledged by the Authors too.

   This information will be added in the new version of the manuscript.

8. At the end of section 4.4, Authors state that "thresholds identified for the Lisboa-Geofisico rain gauge can be applied with reasonable confidence for an area within 50 km distance". This sentence is very difficult to justify, and I think that the ratio a/b or the ratio FN/(TP+FN) cannot be used to justify that assumption. In my opinion, the number of TP and FN are related to the efficiency of prediction model rather than to the representativeness of the rain gauge. I suggest to better justify that sentence or to modify it.

   We acknowledge the comment of the reviewer.
   In fact, we didn't consider the ratio a/b to evaluate the regional performance of the lower limit threshold. We agree that the ratio FN/(TP+FN) better express the efficiency of prediction model. In the new version of the manuscript the end of section 4.4 will be rephrased as follows:
   "Therefore, taking into account the ratio FN/(TP+FN) we can conclude that the prediction model remains efficient up to 50 km distance from the rain gauge. Consequently, although established with landslide data registered up to 10 km distance, the thresholds identified for the Lisboa-Geofísico rain gauge may be applied with reasonable confidence for the area within 50 km distance."

9. In the discussion, a section/paragraph with a brief discussion about the possible application of the thresholds in an operational or prototypal landslide early warning system could introduce more appeal to the work, also in the perspective of the topic of the Special Issue. I list some

works that could be mentioned at this regards: Aleotti (2004); Tiranti and Rabuffetti (2010); Segoni et al. (2015); Calvello et al. (2015); Piciullo et al. (2017).

We acknowledge the importance of the landslide early warning system (lews). However, in fact, the subject of this work is only on the basis of a lews, and because of that we consider inappropriate to open a sub-section on the subject in the discussion section, as the subject was not addressed in the methodology and results sections. Anyway, we have addressed the topic in the conclusion, and we will enlarge it in the new version of the manuscript.

10. I have also some comments about some words used in the text. First, I would suggest using the word "method" (i.e., a systematic way of doing something, implying an orderly logical arrangement, usually in steps) instead of the word "methodology" (i.e., a system of methods followed in a particular discipline). Likewise, I would suggest using "type" (a subdivision of a particular kind of thing) instead of "typology" (a classification according to general type).

The changes in terminology will take into account the reviewer suggestion.

11. Moreover, according to the words "precipitation" and "rainfall", I would ask if the Authors know the type of data recorded by the Lisbon rain gauge. If it includes all types of precipitation (i.e., also snow) it is fair to use the term "precipitation"; otherwise I would suggest using simply "rainfall".

We acknowledge the comment of the reviewer. In Lisboa-Geofísico rain gauge the data recorded is mainly rainfall. Therefore, the term "precipitation" will be changed to "rainfall" in the new version of the manuscript.

12. Finally, I am somewhat dubious about using the word "quantity", as for "rainfall quantity". I would suggest the most used "cumulated rainfall".

We will follow the reviewer suggestion, therefore, the "rainfall quantity" will be changed to "cumulated rainfall".

Moreover, some specific comments are listed below.

13. Page 2, Line 4: I suggest to add "and variables" after "rainfall measurements".

The reviewer's suggestion will be made in the new version of the manuscript.

14. Page 2, Line 10: I suggest to change "triggered during periods of rainfall of short durations" into "triggered by short and intense rainfall".

The change will be made in the new version of the manuscript.

15. Page 2, Line 31: Nikolopoulos et al. (2014; 2015) addressed the issue of rain gauge representativeness.

The reviewer is right and the credits to Nipoloulos et al. (2014; 2015) works will be made.

16. Page 4, Lines 3-5: this sentence is quite complicate and hard to follow. Please reword.

The sentence will be slightly changed to state more clearly.

17. Page 5, Line 16: the equation is quite ambiguous. Please clarify it and add an example.

We agree with the comment of the reviewer. In fact, we consider that the computation of the cumulative antecedent precipitation for different number of consecutive days does not need

to be supported by any equation. So, this equation will be removed in the new version of the manuscript.

18. Page 9, Lines 19-20: I suggest to rephrase the sentence "The relationship between critical duration and month: : :" as "The monthly distribution of critical durations". It seems clearer to me.

We acknowledge the comment of the reviewer and the change will be done in the new version of the manuscript.

19. Page 10, Line 3: the sentence "however none of them ensure the false negatives occurrence" is not clear. Do the Authors mean "none of them ensure a low number of FN"?

The change in the sentence will be done as suggested by the reviewer.

20. Page 11, Line 2: does the PPr index indicate probability of landslide occurrence or probability of landslide prediction?

The PPr index indicates the probability of a rainfall event resulting in a landslide event when a particular threshold is exceeded. We believe this is enough clear in the manuscript.

21. Page 14, Lines 11-13: in the cited works, the criteria used to select the representative rain gauges were based not only on the topographic distance between gauge and landslide, but also to elevation difference and morphological settings.

The reviewer is right and this information will be added to the new version of the manuscript.

Tables
22. Tables 3, 4, and 5 could be merged into one figure (by transposing tables 4 and 5 and by merging them to table 3). They have the same number of rows and columns. Moreover, I suggest using only the acronyms (TP, FN, FP, TN, TPr, FPr, FAr, TS, PPr) instead of the entire names of these parameters and indexes. In Table 3, I suggest to use a specific letter for indicating the variable of the rainfall in the threshold equation, e.g. "R" instead of "y".

We agree with the suggestion of the reviewer and the Tables 3, 4 and 5 will be merged.

Figures
23. Figure 2. I would suggest adding a moving average, other than the MAP line. The red dots represent only the landslides in the 10 km buffer?

We use Figure 2 to demonstrate the inter-annual variability of rainfall together with the temporal distribution of landslide events. As the dry season lasts 3-4 consecutive months each year in the study area, rainfall characteristics of any year does not reflects in the following years concerning landslide activity. Because of this, the association of data to a rainfall yearly moving average is not recommended. Therefore, we think more appropriate to maintain the MAP line.
The dots represent only the landslides in the 10 km buffer. We will add this information to the title of figure 2 to clarify this point. In addition, this figure will also include the non-rainfall triggered landslide events.

24. Figure 3. In order to avoid misunderstanding, I suggest using another shape or another colour to symbolize the 30-day cumulative antecedent rainfall, instead of the red dots (used in figure 2 to represent the landslides). Please correct "Dez" and "Mai" in the labels of x-axes.

The changes will be done as suggested by the reviewer. There were spelling mistakes in the month abbreviations that will be corrected.

25. Some technical corrections:
    Page 3, Line 20: delete the space after the open bracket.
    Page 5, Line 16: add a "+" before ": : :" in the equation.
    Page 9, Line 14: delete "the" before "Fig. 3".
    Page 9, Line 16: correct "70 percentile" and "90 percentile". I suggest to use "70th" or "70%".
    Page 9, Line 27: delete "the" before "Fig. 6".
    Page 9 and followings (also in Table 3): I suggest to use a specific letter for indicating the variable of the rainfall in the threshold equation, e.g. "R" instead of "y".

The corrections will be made in the new version of the manuscript.

26. References (some of them are already cited in the manuscript)
    Aleotti, P.: A warning system for rainfall-induced shallow failures, Eng. Geol., 73, 247– 265, doi:10.1016/j.enggeo.2004.01.007, 2004.
    Calvello, M., d'Orsi, R.N., Piciullo, L., Paes, N., Magalhaes, M.A., Lacerda, W.A.: The Rio de Janeiro early warning system for rainfall-induced landslides: analysis of performance for the years 2010–2013, Int. J. Disast. Risk Reduc., 12, 3–15, doi:10.1016/j.ijdrr.2014.10.005, 2015.
    Gariano, S.L., Brunetti, M.T., Iovine, G., Melillo, M., Peruccacci, S., Terranova, O.G., Vennari, C., Guzzetti, F.: Calibration and validation of rainfall thresholds for shallow landslide forecasting in Sicily, southern Italy, Geomorphology, 228, 653–665, doi:10.1007/s11069-014-1129-0, 2015.
    Giannecchini, R., Galanti, Y., D'Amato Avanzi, G.: Critical rainfall thresholds for triggering shallow landslides in the Serchio River Valley (Tuscany, Italy), Nat. Hazards Earth Syst. Sci., 12, 829–842, doi:10.5194/nhess-12-829-2012, 2012.
    Martelloni, G., Segoni, S., Fanti, R., Catani, F.: Rainfall thresholds for the forecasting of landslide occurrence at regional scale, Landslides, 9, 485–495, doi:10.1007/s10346-011-0308-2, 2012.
    Nikolopoulos, E.I., Borga, M., Creutin, J.D., Marra, F.: Estimation of debris flow triggering rainfall: Influence of rain gauge density and interpolation methods, Geomorphology, 243, 40–50, doi:10.1016/j.geomorph.2015.04.028, 2015.
    Nikolopoulos, E.I., Crema, S., Marchi, L., Marra, F., Guzzetti, F., Borga, M.: Impact of uncertainty in rainfall estimation on the identification of rainfall thresholds for debris flow occurrence, Geomorphology, 221, 286-297, doi:10.1016/j.geomorph.2014.06.015, 2014.
    Piciullo, L., Gariano, S.L., Melillo, M., Brunetti, M.T., Peruccacci, S., Guzzetti, F., Calvello, M.: Definition and performance of a threshold-based regional early warning model for rainfall-induced landslides, Landslides, 14, 995-1008, doi: 10.1007/s10346-016-0750-2, 2017.
    Segoni, S., Rossi, G., Rosi, A., Catani, F.: Landslides triggered by rainfall: a semiautomated procedure to define consistent intensity-duration thresholds, Comput. Geosci.,63, 123–131, doi:10.1016/j.cageo.2013.10.009, 2014.
    Segoni, S., Rosi, A., Rossi, G., Catani, F., Casagli, N.: Analysing the relationship between rainfalls and landslides to define a mosaic of triggering thresholds for regional scale warning systems, Nat. Hazards Earth Syst. Sci., 14, 2637–2648, doi:10.5194/nhess-14-2637-2014, 2014.
    Segoni, S., Battistini, A., Rossi, G., Rosi, A., Lagomarsino, D., Catani, F., Moretti, S., Casagli, N.: Technical note: an operational landslide early warning system at regional scale based on space–time variable rainfall thresholds. Nat. Hazards Earth Syst. Sci., 15, 853–861, doi:10.5194/nhess-15-853-2015, 2015.
    Tiranti, D., Rabuffetti, D.: Estimation of rainfall thresholds triggering shallow implementation, Landslides, 7, 471–481, doi:10.1007/s10346-010-0198-8, 2010.

We acknowledge the reviewer comment and main publications will be included in the new version of the manuscript.

---

## Author Comment (AC2) · 26 Jan 2018

Hazards Earth Syst. Sci. Discuss., https://doi.org/10.5194/nhess-2017-362
Regional rainfall thresholds for landslide occurrence using a centenary database, by Teresa Vaz, José Luís Zêzere, Susana Pereira, Sérgio C. Oliveira, Ricardo A. C. Garcia, and Ivânia Quaresma

**Reply to Anonymous Referee #2**

We appreciate the comments given by the reviewer #2. Detailed answers to the reviewer comments are presented next, item by item. The reviewer comments are presented in black, followed by our answer in blue.

1. The paper by Vaz et al. deals with the identification of landslides triggering thresholds. The paper is well written, even if the topic is highly studied and some drawbacks are found. The most critical issue is due to the fact that landslides are not classified in terms of mechanism, material or volume (3.1 paragraph, line 23). I don't know if the authors could provide at least one these, but I think that an effort in at least one of them could affect the final results. In fact, antecedent rainfalls are very important for landslides affecting soils, but they could be less relevant for rockfalls...
   Most of the adopted landslides are located in urban area, where the fall of walls and some artificial cut could have been termed landslides.
   Several other landslides are located close to the sea, affecting the sea cliff, are the authors sure that they were not triggered by waves?

   We appreciate the comments given by the reviewer #2. We will improve the description of the database (subsection "3.1 Identification of landslide events") addressing the points raised by the reviewer. An amended version of the section 3.1 is presented below:

   "Additionally, using the same newspaper sources, landslides that did not caused any human damage during the same time period were identified and included in the database that supported this study. It should be pointed out, that fall of walls and instabilities directly resulting from engineering works were rejected. Similarly, the landslides in active coastal cliffs were not included in the database. The database structure is divided in two sections: landslide features and landslide damages. The first section includes information of landslide type; temporal and spatial location; triggering factor; and newspaper metadata. The second section refers to human consequences of landslides (fatalities, injuries, missing people, evacuated and homeless people), and direct and indirect damage in buildings, structures, roads and railroad. Our analysis is focused on the date of landslide occurrences. So, the newspapers are a reliable data source, despite the existing uncertainty concerning the spatial location of many reported landslide events and their type. Only landslides with at least 1 day of accuracy were included in the database. The spatial accuracy of landslides cases was classified, following Zêzere et al. (2014), in 5 classes: (i) location with exact coordinates (accuracy associated with scale 1:1 000); (ii) location based on local toponymy (accuracy associated with scale 1:10 000); (iii) location based on local geomorphology (accuracy associated with scale 1: 25 000 scale); (iv) location in the centroid of the parish; and (v) location in the centroid of the council. A total of 400 landslide cases were inventoried being the majority (83%) located with accuracy corresponding to classes (i) to (iii). These landslides affected clay (40.24%), sandstone and conglomerate (22.52%), limestone (16.52%), volcanic (11.11%), marly and marly limestone (9.01%) and

granite (0.60%). The landslide type was classified following the Cruden and Varnes (1996) classification scheme. The slides are the dominant landslide type in the database (53.8%), followed by falls (14.4%). Flows and complex slope movements are less representative (2.4% and 1.5%, respectively). The landslide type is frequently unknown (27.9%). In this study the analysis was performed for all landslide types, following the approach of similar studies (e.g. Brunetti et al., 2010; Rosi et al., 2012; Peruccacci et al., 2017)."

2. I understand the attempt to find a filtering criterion in Page 6 lines 15 -19, but as the authors say it is arbitrary and it is in contrast with the analysis they perform within 10 km, where most of them could be anthropic-induced (pag. 8 lines 23-25). Maybe, thresholds could be evaluated in the range 5 - 10 km, excluding urban landslides. All the performed analyses and the considerations carried out are reasonable and well described, but main drawbacks are the input data.

We appreciate the suggestion of the reviewer to perform the threshold in the range 5 - 10 km. To answer this topic we will include in the subsection "5.3 Identification of landslide rainfall-triggered events" the follow paragraph:

"An alternative method, to the 3-year return period criterion could be the calculation of the thresholds in the range 5 - 10 km, thus excluding the current urban area. However, the landslide database used in this analysis covers a very large time period (145 years) and the urban area extension did change considerably. For example, at the end of the 19th century the rural zones were present within the 5 km buffer. Moreover, this option would reduce the number of landslide events considered in the analysis from 96 to 37, which would reduce the reliability of the obtained rainfall thresholds."

3. minor issues:
   pag 2 line 8 change depth with height
   pag 3 line 6 change quantity with quantify
   pag 4 line 3 remove brackets for April etc
   pag 4 line 24 change along with In
   pag 9 line 13 most rainy
   pag 9 line 27 change detaches with identifies
   Fig 1 it is better to categorize the elevation
   Fig 3 some labels are wrong , i.e. Dez

We acknowledge the reviewer corrections, which will be integrated in the new version of the manuscript.